# Predicting input signals of transcription factors in *Escherichia coli*

Julian Trouillon, Alexandra E Huber [ID], Yannik Trabesinger [ID] & Uwe Sauer [ID] ✉

## Abstract

**The activity of bacterial transcription factors (TFs) is typically modulated through direct interactions with small molecules. However, these input signals remain unknown for most TFs, even in well-studied model bacteria. Identifying these signals typically requires tedious experiments for each TF. Here, we develop a systematic workflow for the identification of TF input signals in bacteria based on metabolomics and transcriptomics data. We inferred the activity of 173 TFs from published transcriptomics data and determined the abundance of 279 metabolites across 40 matched experimental conditions in *Escherichia coli*. By correlating TF activities with metabolite abundances, we successfully identified previously known TF–metabolite interactions and predicted novel TF effector metabolites for 41 TFs. To validate our predictions, we conducted in vitro assays and confirmed a predicted effector metabolite for LeuO. As a result, we established a network of 80 regulatory interactions between 71 metabolites and 41 *E. coli* TFs. This network includes 76 novel interactions that encompass a diverse range of chemical classes and regulatory patterns, bringing us closer to a comprehensive TF regulatory network in *E. coli*.**

**Keywords** Regulatory Network; Metabolite–Protein Interaction; Transcription Factor Activity; Transcriptomics; Metabolomics
**Subject Categories** Chromatin, Transcription & Genomics; Metabolism; Microbiology, Virology & Host Pathogen Interaction

## Introduction

Adapting to changing environmental conditions requires adjustments in the cellular proteome. In bacteria, these adjustments are primarily mediated by transcription factors (TFs), which regulate gene expression through their interactions with DNA (Browning and Busby, 2016; Mejía-Almonte et al, 2020; Fang et al, 2017). To elicit appropriate gene expression responses, information from the extracellular environment or the cellular state must be relayed to TFs. The predominant mechanism for this information transfer in bacteria is the allosteric binding of intracellular metabolites (Ledezma-Tejeida et al, 2021; Yugi and Kuroda, 2018; Femerling et al, 2022; Habibpour et al, 2024). Most bacterial TFs contain small-molecule binding domains that enable them to directly sense these intracellular input signals (Ledezma-Tejeida et al, 2021), which may include intermediates of metabolic pathways or imported metabolites. Upon sensing their effector molecules, TFs undergo conformational changes that influence their DNA-binding capacity—either enhancing or inhibiting it— thereby affecting the expression of their target genes (Browning et al, 2019).

Since the foundational discoveries regarding bacterial TF signal-sensing mechanisms several decades ago (Jacob and Monod, 1961; Hillen et al, 1983; Lewis et al, 1996; Gilbert and Müller-Hill, 1966), extensive research has characterized the regulatory interactions between metabolites and TFs, which serve as input to the gene regulation network, particularly in the model organism *E. coli* (Santos-Zavaleta et al, 2019; Femerling et al, 2022; Browning et al, 2019). As of today, input signals have been identified for about 40% of *E. coli*'s TFs that are predicted to directly sense metabolites (Ledezma-Tejeida et al, 2021; Femerling et al, 2022). However, this proportion decreases dramatically to below 1–2% in non-model organisms (Dudek and Jahn, 2022). The relatively slow pace of the discovery process is largely due to the time-consuming and low-throughput nature of experiments to identify such interactions. Typically, a TF is first characterized by the genes and cellular functions it regulates. The search for input signals generally occurs only after extensive studies that map out the TF targets and characterize their functions, usually by hypothesizing potential signal metabolites from the identified cellular function. These hypotheses are then tested individually using in vitro methods, such as DNA-binding assays, to elucidate the regulatory effects of these interactions (Ricca et al, 1989; Urano et al, 2015; Martí-Arbona et al, 2012; Nemoto et al, 2012; Lim et al, 1987; Quail et al, 1994; Hart and Blumenthal, 2011).

Recent advances in high-throughput methods now enable the measurement of gene expression and metabolite abundances at large scales, significantly accelerating the identification of TF–metabolite interactions (Holbrook-Smith et al, 2024; Orsak et al, 2012; Liu et al, 2020; Huang et al, 2018). One effective approach involves correlating in vivo TF activities with metabolite abundances. The underlying assumption is that changes in the abundance of an input metabolite for a specific TF should correspond with that TF's regulatory activity. Although recent pioneering studies have demonstrated this correlation (Kochanowski et al, 2017; Lempp et al, 2019; Yogendra and Kushalappa, 2016), such applications have thus far been limited to relatively small numbers of TFs or specific growth conditions. Here, we implement a systematic workflow to predict TF input signals in *E.*

Institute of Molecular Systems Biology, ETH Zürich, Zürich 8093, Switzerland. ✉E-mail: sauer@ethz.ch

*coli* by combining the large, publicly available PRECISE2.0 transcriptomics dataset (Rychel et al, 2021) with high-throughput metabolomics. By quantifying the intracellular metabolome across 40 growth conditions that matched those of the transcriptomics dataset, we obtained paired profiles of gene expression and metabolite abundances across a diverse range of growth conditions. After inferring TF activities from gene expression by leveraging the known *E. coli* regulatory network, we identified correlating pairs of TFs and metabolites, successfully recovering known interactions and revealing novel ones.

# Results

## Inferring transcription factors activity from gene expression

To determine TFs activities, we leveraged published transcriptomics data from the PRECISE2.0 *E. coli* dataset that covers approximately 400 growth conditions (Fig. 1A,B) (Rychel et al, 2021), as well as known TF target genes from the RegulonDB regulatory network (Fig. 1C; Dataset EV1) (Santos-Zavaleta et al, 2019). For each TF, activity was defined as the functional influence that a TF exerts on the expression of its direct target genes, inferred from the collective expression pattern of those targets in its regulon. We tested six different computational methods to assess their ability to correctly predict expected decreases of TF activity in mutant strains from the PRECISE2.0 dataset, where a given TF had been knocked out. Among these, the VIPER method (Alvarez et al, 2016) yielded the best results, correctly assigning 34 out of 40 pairs (Fig. EV1A–C, "Methods") and was selected for further analysis. Subsequently, we evaluated whether the inferred TF activities corresponded with the presence of TF input signals. The PRECISE2.0 dataset includes 47 conditions wherein at least one metabolite known to interact with a TF was added to the growth medium, which comprises known interactions for 23 TFs. In 83% of these cases, we observed the expected direction of TF activity change when compared to a paired control condition with no metabolite added, i.e., increased or decreased TF activity in the presence of corresponding activating or inactivating signal molecule, respectively (Fig. 1D). For the remaining 17%, the activity changes were relatively low, suggesting little effect of the addition of the metabolite, which could be due to low metabolite uptake, fast metabolite degradation, or inactivity of the corresponding TF under the specific condition. Altogether, the expected changes were evident at both the TF activity and target gene expression levels, as illustrated for four TFs exhibiting diverse combinations of activating/inactivating TF–metabolite interactions and activating/repressing TF–gene interactions (Fig. 1E,F). Based on these analyses, we conclude that the inferred TF activities are biologically meaningful and can be effectively used to predict TF input signals.

## Selecting a set of reproducible growth conditions

To systematically identify novel TF input signals, we aimed to generate metabolomics data under conditions where TF activities had been inferred, with the goal of finding metabolite abundances that correlate with these TF activities. Specifically, we searched for a highly diverse set of growth conditions to reduce the number of metabolites with similar abundance profiles, which hinder the identification of signal molecules (Lempp et al, 2019). Approximately 75% of the growth conditions in the PRECISE2.0 dataset pertain to either strains specifically evolved towards phenotypes of interest, strains expressing heterologous DNA, or undefined growth media (Fig. 1A). From the remaining 100 conditions, we successfully reproduced the growth condition for 10 strains in 40 cases (Dataset EV2), which will henceforth be referred to as the selected experimental conditions. These selected conditions encompass a diverse array of nutritional and genetic perturbations, including 8 TF knockout mutants (Fig. 1B).

For the entire PRECISE2.0 dataset, we obtained a wide range of activities for 173 TFs, highlighting the richness of the transcriptomics dataset (Fig. 2A; Dataset EV3). To assess the extent of variability lost by selecting this subset of 40 experimental conditions, we determined the maximum ranges of TF activities; specifically, we quantified the difference between the highest and lowest activity values for each TF across conditions. A higher range value is anticipated to enhance the likelihood of detecting meaningful patterns in the data, as illustrated by the significant increase in correctly assigned TF activities when applying a threshold for higher activity ranges (Fig. EV1E,G). Overall, the 40 selected experimental conditions, representing about 10% of the PRECISE2.0 dataset, retained an average of 72% of the maximum range values across the entire dataset for the 173 measured TFs (Fig. 2B; Dataset EV4), and exhibited a similar pattern of recovery of known interactions as found across all conditions (Figs. 2C and 1D).

## Systematic prediction of TF–metabolite interactions

To identify novel TF input signals, we cultured *E. coli* wild-type MG1655 and BW25113, as well as eight mutant strains under different growth conditions, corresponding to the 40 selected PRECISE2.0 experimental conditions (Dataset EV2). Intracellular metabolites were extracted during the mid-exponential growth phase, and their abundances were determined using untargeted direct flow-injection mass spectrometry metabolomics (Fig. 3A) (Fuhrer et al, 2011). As expected, metabolites that were added in specific growth conditions—including known TF signal molecules—exhibited the highest abundance in their corresponding condition (Fig. 3B). By utilizing the KEGG database (Kanehisa et al, 2017), we annotated 279 metabolites based on their exact masses, for which we obtained relative abundances across a wide range of intensities throughout the 40 selected conditions (Fig. 3C; Datasets EV5 and EV6). To assess the diversity of metabolic states across these 40 selected conditions, we calculated the maximum abundance fold change for each metabolite. For the 279 annotated metabolites, we found a median abundance fold change of 3.34 (Fig. 3C), indicating significant variation for most compounds and metabolic pathways across the selected conditions. This allowed us to generate a diverse metabolomics dataset aligned with the corresponding published transcriptomics dataset. By correlating metabolite abundances with TFs activities, we were able to generate hypotheses regarding potential metabolic input signals of TFs (Fig. 4A).

To ensure the accuracy of our approach and mitigate potential biases stemming from single outliers or poorly matched growth

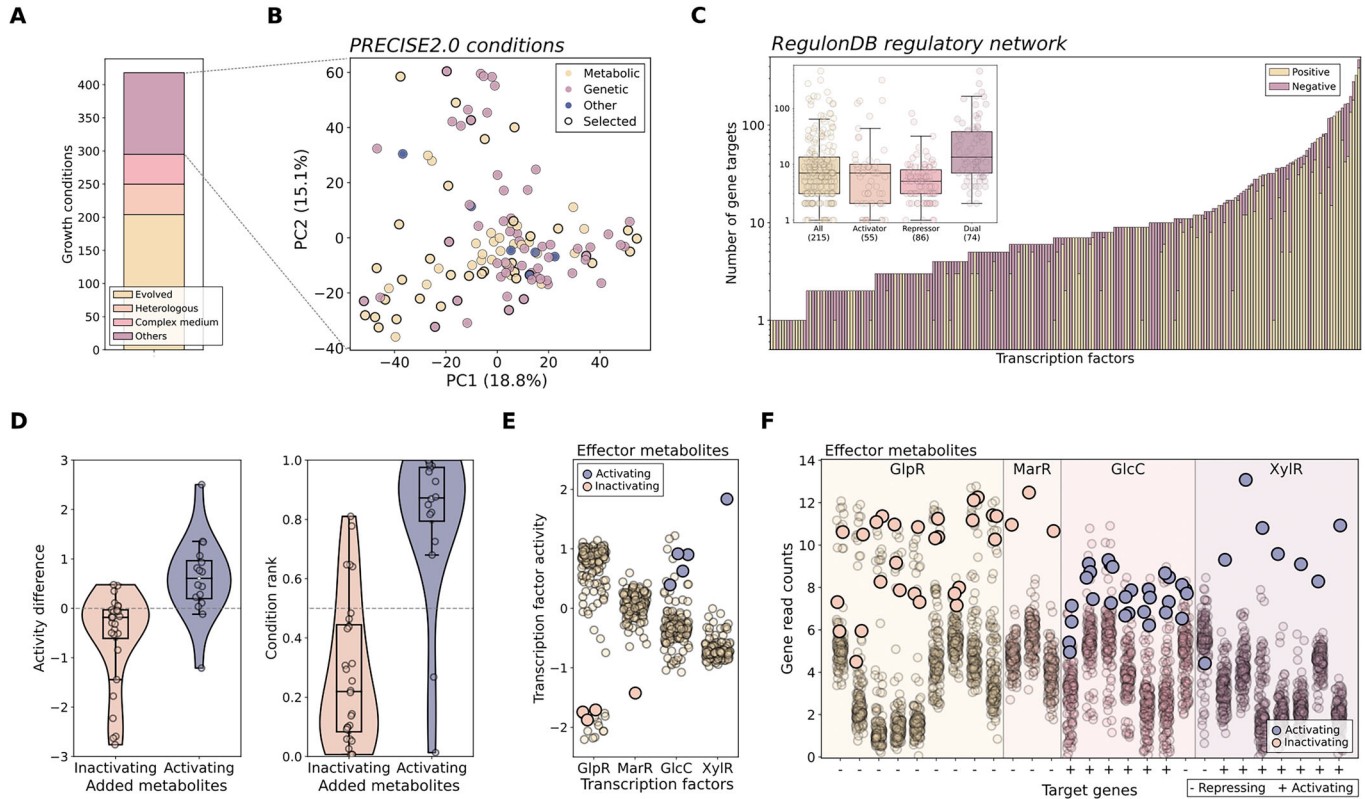

**Figure 1. TF activities inferred from gene expression capture the expected effects of TF input signals.**

(A) Distribution of various types of experimental conditions within the iModulon *E. coli* PRECISE2.0 dataset. "Evolved": condition with strain from an adaptive laboratory evolution experiment. "Heterologous": condition with strain carrying heterologous DNA. (B) Principal component analysis of candidate experimental conditions within the PRECISE2.0 transcriptomics data, with the conditions selected for this study highlighted with black circles. Perturbation types are color-coded as indicated. (C) Enumeration and categorization of regulatory interactions associated with the target genes of all transcription factors, as documented in RegulonDB. (D) Impact of adding a ligand to the medium on the activity of the ligand-regulated TF. The results are shown for all conditions in the PRECISE2.0 dataset where a known TF effector metabolite was introduced ($n = 47$; 29 for inactivating and 18 for activating metabolites). This includes both the activity difference (left) between the condition with the added metabolite and its corresponding control conditions, as well as the rank of the condition with the added metabolite compared to all other conditions (right). (E, F) Influence of ligand addition on TF activity (E) and on the expression of target genes (F) for GlpR, MarR, GlcC, and XylR across all conditions. Conditions involving effectors known for either activating or inhibiting TF activities are indicated in blue or pink, respectively. The directionality of known TF regulations for each target gene is denoted as '−' for repressive regulation and '+' for activating regulation. Source data are available online for this figure.

conditions between the two datasets, we used bootstrapping techniques on the selected conditions for correlation. Specifically, we randomly subsampled 30 out of the 40 selected conditions across 1000 samples and assessed the performance of our correlation approach for each sample. Notably, most previously known TF–metabolite interactions were amongst the top correlating pairs (Fig. 4B,C), illustrating the efficacy of our approach in identifying TFs input signals. Interactions that increase TF activity were particularly well-recovered with an area under the receiver operating characteristic (ROC) curve of 0.81. Conversely, the recovery of interactions that decrease TF activity showed lower performance in negatively correlating pairs at low false positive rates (Fig. 4B); thus, we focused on activating interactions moving forward. In total, we obtained average correlation values for 48,267 TF–metabolite pairs across the 173 TFs (Fig. 4C,D), with highly correlating pairs suggesting numerous potential new interactions.

To generate a list of higher-confidence candidate TF input signals, we implemented three filtering methods to assess the

likelihood of TF–metabolite pairs being involved in regulatory interactions (Fig. EV2). First, we applied a false positive rate threshold of 0.1 based on the median ROC curves. Second, we established a stability metric that indicates, for each pair, the proportion of bootstrapping samples in which the correlation score exceeds the false positive rate threshold. The stability threshold was then defined as the value that maximizes the retention of known TF–metabolite interactions. Third, we defined a distance metric representing the number of consecutive reactions in the metabolic network needed to connect the metabolite involved to the nearest enzyme directly regulated by the TF. Setting a threshold for lower distance values increases the likelihood of identifying functional pairs (Fig. EV2C) (Lempp et al, 2019), because TF input signals typically originate from within or near the regulated pathways (Santos-Zavaleta et al, 2019). However, this filtering restricted our search to TFs with at least one regulated enzyme, which included 132 out of the 173 TFs with inferred activities. For positively correlating pairs, applying the stability and distance filters increased the proportion of recovered known interactions about

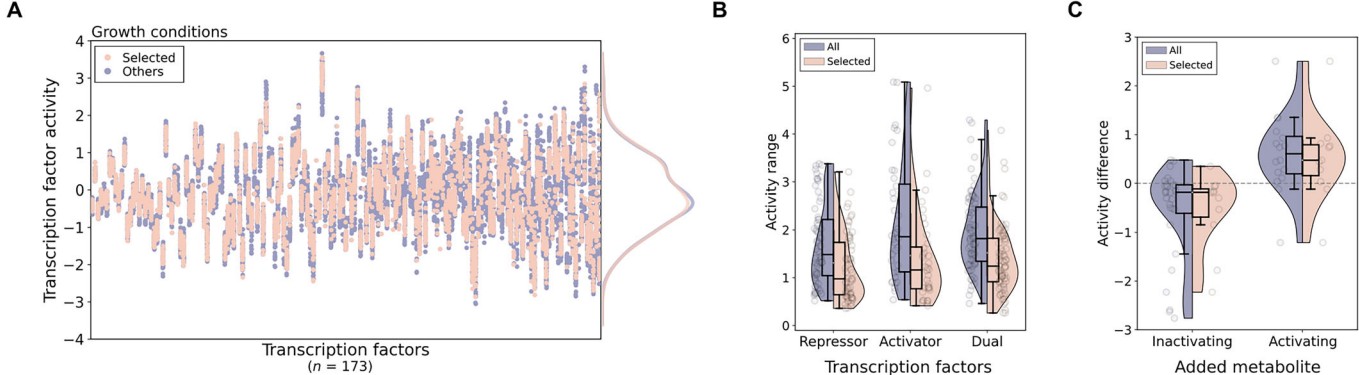

**Figure 2. Selected growth conditions preserve data variability.**

(A) Activities of TFs across all experimental conditions in the PRECISE2.0 dataset, encompassing all TFs in RegulonDB with more than three known target genes. The conditions selected for metabolomics measurements are highlighted in orange, while non-selected conditions are denoted in blue. On the right, the density of overall TF activities is illustrated for both selected (orange) and all conditions (blue). TFs are sorted by activity range. (B) Activity ranges (difference between highest and lowest activity values) of TFs ($n = 173$) across all conditions or only the ones selected for this study. TFs are categorized according to their type of gene regulation: repressors, activators, or dual regulators. (C) Activity differences for TFs with known signals across all PRECISE2.0 conditions or only the ones selected ($n = 47$; 29 for inactivating and 18 for activating metabolites). In all cases, the signal metabolite was added to the corresponding control condition. Source data are available online for this figure.

100-fold when compared to all pairs that passed the false positive rate threshold in at least one bootstrapping sample, resulting in a final list of 103 TF–metabolite pairs (Fig. 5A; Dataset EV7). These 103 pairs all represent high-confidence predictions which cannot be narrowed down further without decreasing the performance of our approach. Among these, we recovered the previously known activating signal molecule for five out of eight transcription factors. Overall, this approach identified candidate input signals for 42 TFs.

Among the 42 TFs with predictions, 18 had previously known small-molecule input signals, including eight that enhanced TF activity and were detected by our metabolomics analysis (Dataset EV7). Our approach correctly predicted the known signal molecules in five out of these eight cases (Fig. 4E). For the remaining three cases, the predicted metabolite was found directly downstream of the known signal, being the product of an enzymatic reaction in which the known signal serves as the substrate. A potential explanation for this could be the rapid conversion of the actual signal molecule during sample processing. Altogether, our approach successfully recovered five of the eight known signal molecules for both direct and indirect TF activations, while in the other cases, it predicted a metabolite one reaction step removed from the signal.

Among the predicted pairs, Crp was the TF with the highest number of candidate effectors, totaling 23 candidates, while the average across TFs was below two (Dataset EV7). With over 600 known targets, Crp possesses the largest array of target genes among all *E. coli* TFs, including hundreds of enzymes. Consequently, many metabolites are in close proximity to Crp targets, which diminishes the effectiveness of our distance filtering process. In addition, the number of predictions was inflated due to the inability of our metabolomic approach to distinguish several metabolites with the same mass, particularly among the predicted sugar inputs to Crp. Despite the relatively high number of candidates generated for Crp, its known effector, cyclic AMP, was among the predicted candidates and had the highest correlation and stability scores.

Beyond Crp, we predicted 80 candidate effector metabolites for 41 TFs. These predicted pairs represent all three established regulatory patterns regarding the position of the signal molecule relative to the regulated enzymatic reactions (Santos-Zavaleta et al, 2019) (Fig. 5B), as follows: Upstream: the metabolite is the first substrate in a metabolic reaction, or a series of reactions regulated by the TF. Within: the metabolite is an intermediate in a chain of reactions regulated by the TF. Downstream: the metabolite is an end product of a reaction, or a series of reactions, regulated by the TF. Novel pairs were predicted for each regulatory pattern with the majority of pairs being from the within pattern (Fig. 5B), which is consistent with known interactions (Santos-Zavaleta et al, 2019). 12 TFs had upstream metabolites identified as input signals, indicating feedforward sensing of the substrates of the regulated reactions. For example, L-glutamine was predicted to be the activating signal of GadX. GadX controls the acid resistance response notably with the activation of *glsA* (Tucker et al, 2003; Tramonti et al, 2008), which is involved in acid resistance through the conversion of L-glutamine to L-glutamate and ammonia (Lu et al, 2013). As a predicted activator of GadX, L-glutamine thus promotes the positive regulation of that pathway (Fig. 5C). Nine cases involved metabolites that were downstream of target enzymes. An example is the repressor MngR, which inhibits two proteins engaged in uptake and cleavage of 2-O-alpha-mannosyl-D-glycerate (Fig. 5D) (Sampaio et al, 2004). In this instance, both cleavage products, mannose-6-P and glycerate, were predicted to activate the inhibiting TF, representing a negative feedback loop. Lastly, 27 TFs were associated with predicted signals that correspond to metabolites within their regulated pathways, such as RutR (Fig. 5E), which regulates pyrimidines metabolism (Shimada et al, 2008) and has two previously known inactivating input signals: uracil and thymine (Shimada et al, 2007). Aminoacrylate and 3-oxopropanoate were both predicted as activating signals of RutR from within the RutR-repressed uracil degradation pathway (Fig. 5E). While uracil induces upregulation of the pathway from upstream through the inactivation of RutR, the two predicted

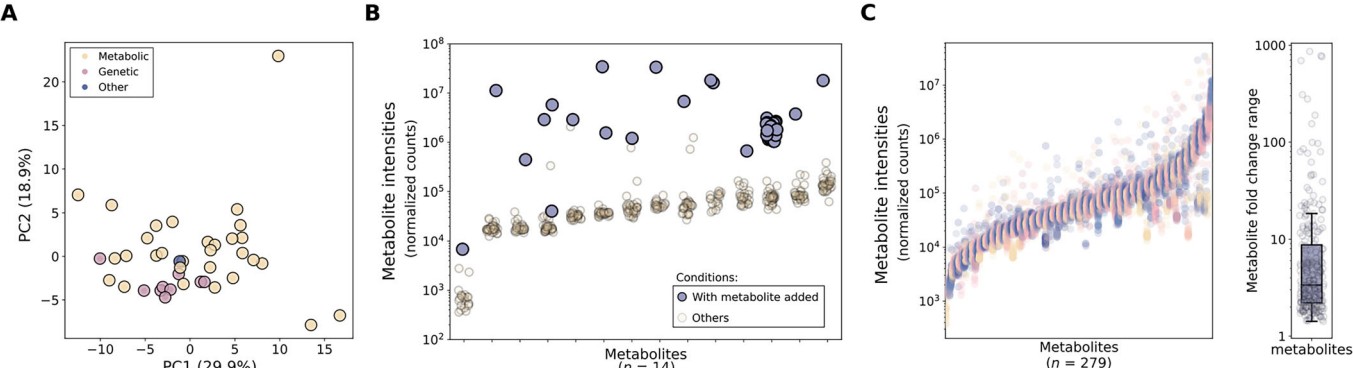

**Figure 3. High-throughput metabolomics probes wide ranges of metabolite abundances across diverse growth conditions.**

(A) Principal component analysis of metabolomics data obtained from the 40 selected experimental conditions. Perturbation types are color-coded as indicated in the legend. (B) Metabolite intensities for 14 metabolites (*x* axis) that have been added to the growth medium in at least one experimental condition are displayed across all experimental conditions. For each metabolite, the measured intensities are represented by individual dots, with blue dots marking conditions in which the respective metabolite was added. (C) Left: All annotated metabolite intensities ($n = 279$) are showcased across all 40 measured conditions with different colors representing different metabolites. Right: Distribution of maximum abundance ranges for all annotated metabolites ($n = 279$) across all growth conditions. Source data are available online for this figure.

signals located after the branching out from carbamate represent negative feedback loops to inhibit the pathway from within that end branch. This negative regulation of the pathway could be needed in case of the accumulation of 3-oxopropanoate, which is toxic (Parales and Ingraham, 2010). Overall, we identified 76 novel candidate input signals spanning all previously known regulatory logics.

## Validation of predicted input signal for LeuO

For some TFs, we predicted multiple plausible input signals with varying regulatory logics. A notable example is LeuO, the activator of the leucine biosynthesis operon that is also a global antagonist against the universal silencer of stress-response genes H-NS (Shimada et al, 2011; Sánchez-Popoca et al, 2022), for which no input signal was previously known (Fragel et al, 2019; Santos-Zavaleta et al, 2019). We predicted four LeuO input signals derived from either upstream or within the leucine biosynthesis pathway (Fig. 6A). The prediction of these multiple input signals may stem from consecutive metabolites displaying correlated abundances, leading to similarly high scores in our analysis. In such scenarios, experimental validation is crucial to pinpoint the exact signal. We chose LeuO for this validation to determine whether one of its four predicted signals influences its DNA-binding function. The LeuO protein was purified and its binding to DNA assessed at the promoter region of one of its target operons, *leuLABCD*, using an electromobility shift assay. To maximize the dynamic range of our assay for measuring the effect of an added molecule on LeuO's DNA-binding activity (either up or down), we first established a LeuO concentration where approximately half of the DNA fragments were bound by the TF, as determined by a concentration gradient (Fig. EV3). Using these conditions, we tested the effect of each of the four predicted input signals on LeuO activity and found that only 2-isopropylmalate significantly enhanced LeuO binding to its target DNA (Figs. 6B and EV4). To further confirm this novel interaction, we employed a thermal shift assay to assess the

influence of 2-isopropylmalate on the stability of the LeuO-DNA complex. Typically, an input molecule modifies the conformation of its bound TF, which in turn impacts its stability. In this assay, 2-isopropylmalate significantly increased the thermal stability of LeuO by 1.6 °C (Fig. 6C), thereby further validating 2-isopropylmalate as the input signal to LeuO.

## Discussion

Even in *E. coli*, arguably the best-studied bacterium, we still do not know the signals that most TFs sense to initiate transcriptional responses (Femerling et al, 2022; Ledezma-Tejeida et al, 2021). This knowledge gap can primarily be attributed to the labor-intensive and low-throughput methodologies that have been employed to elucidate these interactions. Using paired transcriptomics and metabolomics, we correlated the regulatory activities of TFs with the abundance of metabolites to predict signal molecules across 173 TFs. Our approach recovered most known activating TF–metabolite interactions, among the detected metabolites, but also unveiled numerous novel interactions. Upon refining our predictions, we identified a high-confidence ensemble of predicted TF–metabolite interactions encompassing 80 metabolites and 41 TFs. These predicted signal molecules span a wide range of chemical classes and regulatory patterns, including both metabolic feedforward and feedback regulations. The outcome of this study proposes a reproducible experimental pipeline to identify TFs input signal with unprecedented throughput. This advancement identified numerous novel regulatory interactions and brought us closer to a comprehensive transcriptional regulatory network in *E. coli*.

For experimental validation, we focused on LeuO, a TF that regulates diverse cellular processes in gammaproteobacteria (Hernández-Lucas and Calva, 2012), including its most conserved target: the adjacent leucine biosynthesis operon *leuABCD*. As a LysR-like TF, LeuO features a predicted metabolite-binding domain. Notably, mutations that mimic a bound state enhance

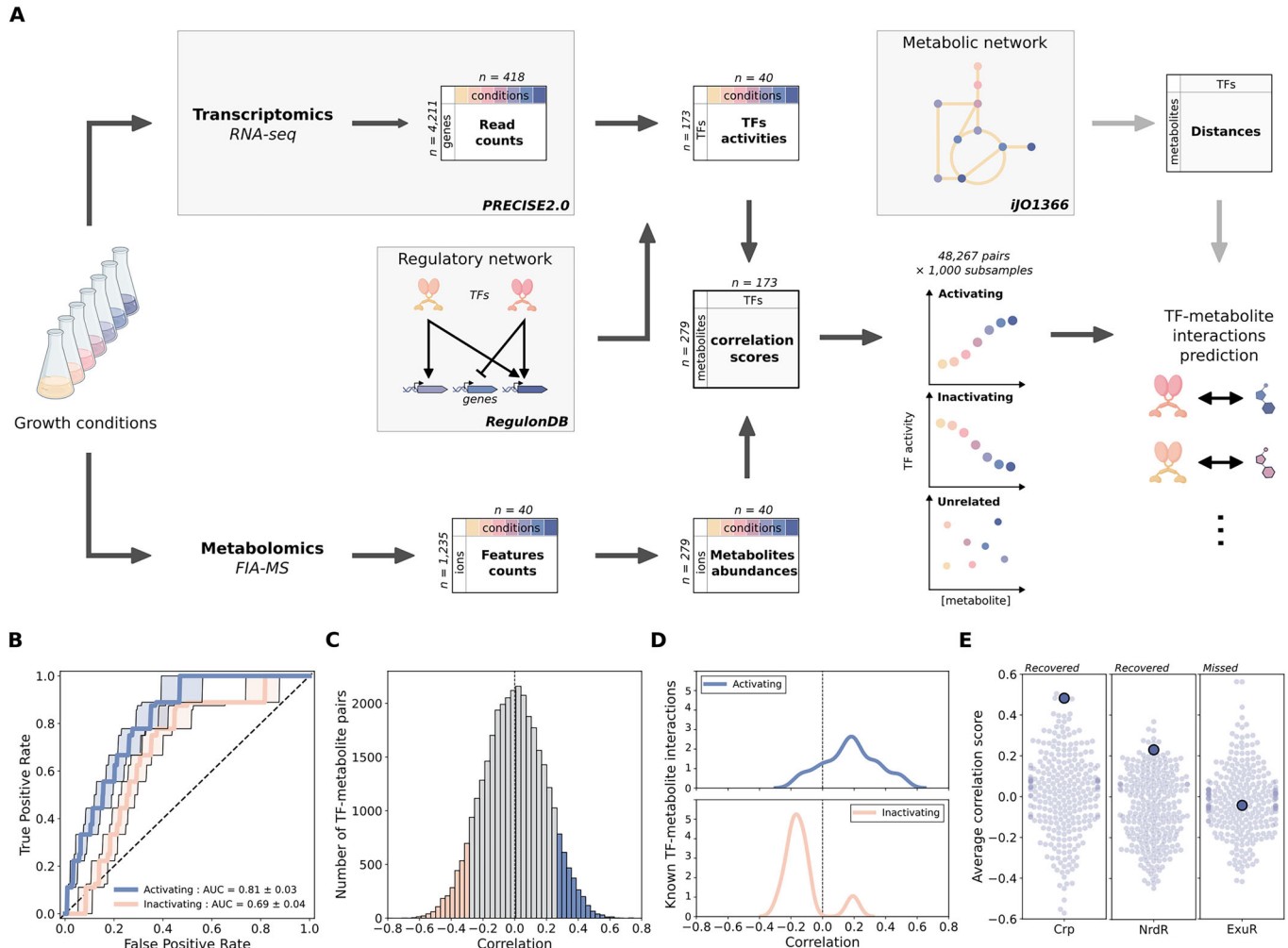

**Figure 4. Correlation analysis of TF activities and metabolite abundances recover previously known TF–metabolite interactions.**

(A) Schematic representation of the approach used in this study. RNA-seq data and the regulatory network were obtained from the iModulon and RegulonDB databases, respectively. We replicated 40 growth conditions based on the iModulon PRECISE2.0 dataset to conduct metabolomics analyses, thereby obtaining paired transcriptomics and metabolomics datasets. TF activities were inferred from the transcriptomics data and subsequently analyzed in conjunction with metabolite abundances through correlation analysis. (B) Median receiver operating characteristic curves are illustrating, showcasing the recovery of known activating (blue) and inactivating (pink) TF–signal interactions. (C) Distribution of correlation scores across all TF–metabolite pairs, with bins either above or below the false positive rate of 0.1 correlation thresholds in both directions highlighted in color. (D) Distribution of correlation scores for known activating (top) or inactivating (bottom) TF–signal interactions, illustrating the prevalence and strength of these relationships. (E) Distribution of TF–metabolite correlation scores for Crp, NrdR, and ExuR with all measured metabolites (individual dots). The respective known signal molecules are highlighted as large blue dots. The known interactions of Crp and NrdR were correctly recovered by our approach, while the one of ExuR was missed. Source data are available online for this figure.

LeuO's regulatory activity; however, the input signal remained unidentified (Fragel et al, 2019). We experimentally validated the predicted 2-isopropylmalate as the signal metabolite that increases LeuO activity upon binding. As an intermediate in leucine biosynthesis, 2-isopropylmalate is the first pathway-specific metabolite to emerge after the divergence from valine biosynthesis (Calvo et al, 1962). Our findings suggest a positive feedforward loop whereby LeuO activation promotes leucine biosynthesis in response to the accumulation of this initial pathway-specific metabolite. While our results clarify regulation of the leucine pathway, the regulatory logic by which LeuO governs processes beyond leucine synthesis remain uncertain, especially considering its more than 100 gene targets (Dillon et al, 2012; Shimada et al, 2011;

Hernández-Lucas and Calva, 2012). This scenario bears resemblance to other broadly acting TFs, such as Lrp, PdhR or ArgP, which also respond to a single biosynthesis pathway-specific metabolite (Kroner et al, 2019; Anzai et al, 2020; Nguyen Le Minh et al, 2018). In these cases, the pathway related to the input signals are typically well conserved among the target genes, while additional targets display variability across species or strains, indicating their later incorporation into the TF regulon (Trouillon et al, 2020; Baumgart et al, 2021).

Among all methods tested for TF activity inference, VIPER was the best-performing method on our dataset. To apply our approach to other bacterial species, VIPER can be directly used without the need for TF mutant strains, which were used solely for

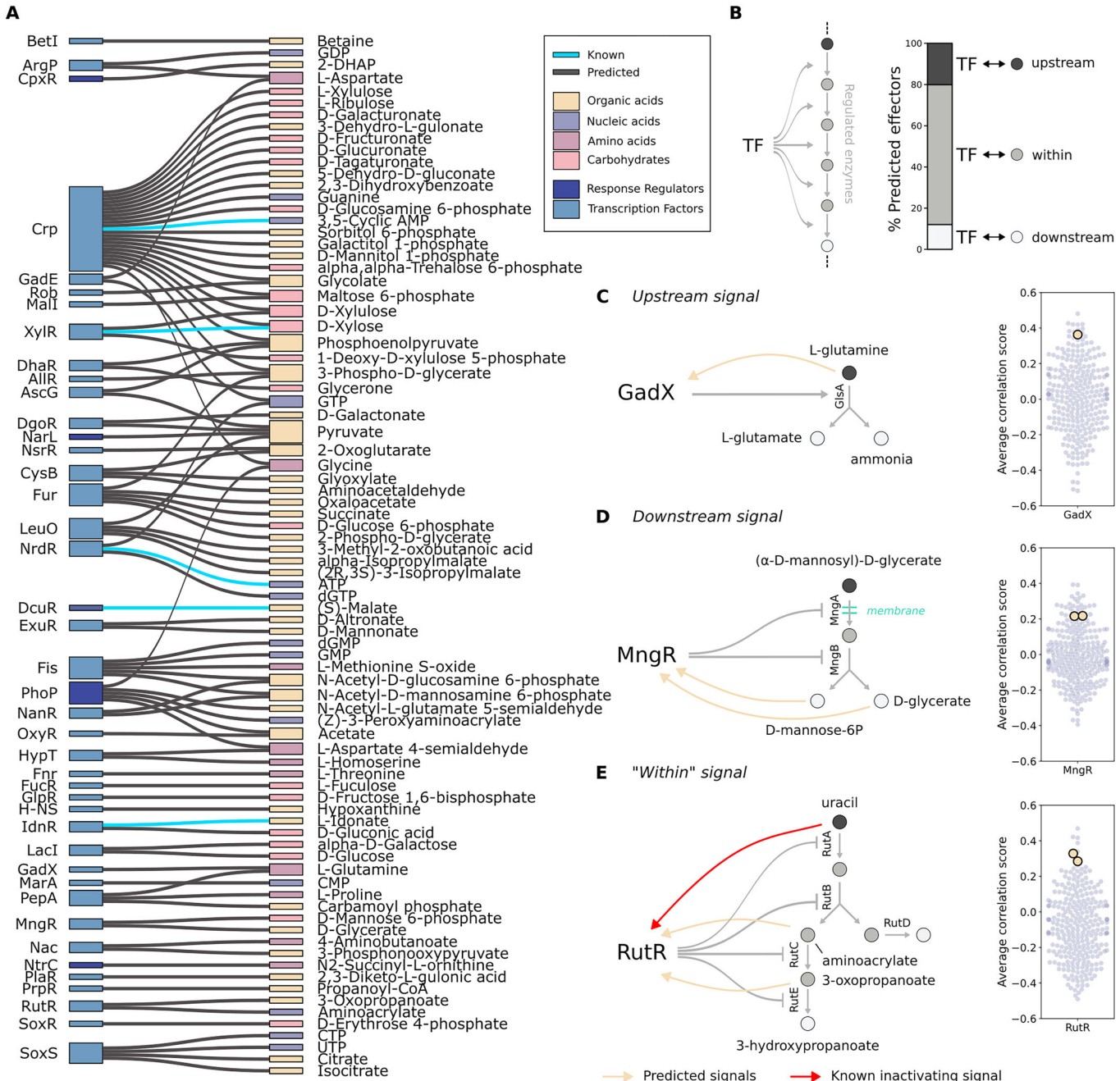

**Figure 5. Predicted input signals encompass a wide range of chemical classes and regulatory logics.**

(A) Sankey diagram illustrating the predicted pairs of TFs and signals. TFs and metabolites are color-coded according to various categories, as specified in the graphical legend. Light blue links indicate previously known activating TF–metabolite interactions. (B) Proportions of predicted signals based on their location within the metabolic sub-network that is directly regulated by the respective TF. (C–E) Example predictions for upstream (C), downstream (D), or within (E) predicted metabolic input signals. Left: Schematic representation of predictions. Predicted TF–signal interactions are represented with yellow arrows, and metabolites are colored as shown in panel b. Regulated enzymatic reactions are depicted in gray to represent their role in the metabolic context. For RutR (E), the previously known inactivating input signal is illustrated with a red arrow. Right: Distribution of TF–metabolite correlation scores for the corresponding TFs with all measured metabolites. The respective predicted signal molecules are shown as large yellow dots. Source data are available online for this figure.

benchmarking different methods. Inferring TF activity is challenging, particularly due to the overlap within TFs regulons, which results in individual genes being regulated by multiple TFs. Inference of TF activity is necessary because the expression levels of TFs are often poor proxies for their actual activity (Larsen et al,

2019). This discrepancy may largely stem from the considerable influence of post-transcriptional regulation. In addition, since it is not established for all TFs whether they regulate their own expression, incorporating TF expression levels when inferring TF activity becomes problematic. The VIPER method relies on a

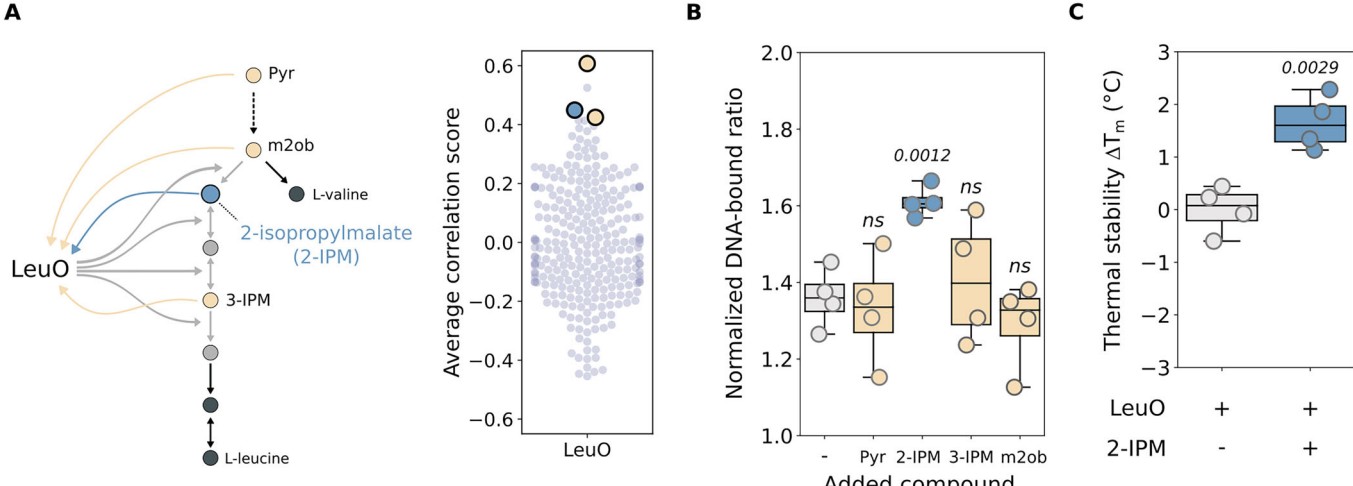

**Figure 6. LeuO responds to 2-isopropylmalate.**

(A) Predicted activating input signals for LeuO. Left: Schematic representation of predictions. Predicted TF–signal interactions are represented by yellow or blue arrows, with the blue arrow representing the signal metabolite 2-isopropylmalate (2-IPM), which is shown to interact with LeuO in (B, C). Metabolites are colored as in Fig. 5b, while regulated enzymatic reactions are depicted in gray. The dotted reaction arrow represents multiple consecutive enzymatic reactions that are not detailed in this scheme. Pyr pyruvate, m2ob 3-methyl-2-oxobutanoate, 3-IPM 3-isopropylmalate. Right: Distribution of TF–metabolite correlation scores for LeuO with all measured metabolites. The predicted signal molecules are shown as large yellow or blue dots, with the blue dot representing the signal metabolite 2-IPM, which is shown to interact with LeuO in (B, C). 2-IPM and 3-IPM are shown as a single dot as they have the same mass. (B) Change in LeuO DNA-binding activity on the *leuLABCD* promoter region upon the addition of predicted signal metabolites as measured by electromobility shift assay. DNA binding is quantified as the ratio of bound to unbound DNA fragments for each gel lane, normalized to a control lane lacking LeuO protein. Data from four independent gel shift experiments are displayed as individual dots. (C) Change in thermal stability of LeuO upon the addition of 2-isopropylmalate (2-IPM) as measured by a thermal shift assay conducted in four independent experiments. Results for each replicate are shown as the difference between its melting temperature and the average melting temperature of the control samples without 2-IPM ($\Delta T_m$). Exact $P$ values below 0.05 from two-tailed $t$ tests are indicated above the corresponding box when compared to the control samples with "ns" denoting non-significant. Source data are available online for this figure.

probabilistic framework to evaluate the enrichment of specific TF regulons among differentially expressed genes. Its use of analytic rank-based enrichment analysis, along with permutation tests for normalization, enhances its robustness. These features make the method resilient to signature noise, regulon subsampling, and variations in sample quality, which likely contribute to its superior performance on this dataset.

As part of our systematic workflow, we used a correlation analysis based on gene expression and metabolite abundance akin to earlier work (Lempp et al, 2019; Kochanowski et al, 2017; Yogendra and Kushalappa, 2016), and applied it to a wide range of experimental conditions. While the approach presented here accelerates the identification of TFs input signals, it has several limitations. Firstly, the activities of TFs are inferred from previously identified gene targets for each TF, which means our approach heavily relies on existing knowledge of the transcriptional network (Badia-i-Mompel et al, 2022). This knowledge base is currently quite incomplete for most organisms (Santos-Zavaleta et al, 2019), even in *E. coli* covering only about 30% of the regulatory network (Trouillon et al, 2023). Consequently, predictions could be made for only 173 out of ~300 predicted TFs in *E. coli*. Expanding the knowledge base not only increases the number of TFs with predictions but would also enhance the quality of these predictions by refining and enlarging the set of TF-regulated targets. Even well-studied TFs often lack comprehensive target annotations or may have reproducibility issues with their annotated targets. Secondly, the selection and number of experimental conditions used for correlation analysis strongly influence the potential for

identification of new interactions (Lamoureux et al, 2023). In a previous study, the analysis was limited to transitions between two growth conditions, which restricted the interpretation of results, as many metabolites exhibited similar abundance dynamics (Lempp et al, 2019). We mitigated this issue by measuring metabolite abundances across 40 diverse experimental conditions, thereby reducing the likelihood of obtaining highly similar profiles. Nevertheless, the specific choice of conditions remains influential, as many TFs are only expressed or active under particular circumstances. Lastly, the number and types of metabolites measured are contingent upon the chosen metabolomics method (Alseekh et al, 2021). By relying on MS1-based annotation, several metabolites could not be distinguished in our analysis, potentially resulting in an artificially inflated number of predicted metabolites in some cases, such as for Crp. To enhance coverage of detected metabolites, the integrated use of multiple metabolomics methods may prove beneficial.

The application of high-throughput inference methods often leads to increased false positive rates compared to targeted in vitro validation experiments (Lempp et al, 2019; Piazza et al, 2018; Liu et al, 2020; Huang et al, 2018; Orsak et al, 2012; Lindsley and Rutter, 2006; Peng et al, 2025). Recently, two other high-throughput approaches identified numerous potential TF–metabolite interactions in *E. coli* (Peng et al, 2025; Lempp et al, 2019). Notably, both sets of predictions shared 11 TFs with our results, among which we successfully recovered the majority of known interactions. However, only two of our predicted interactions overlapped with those from one of these studies (Lempp et al, 2019): NanR

with N-acetyl-D-glucosamine-P and DhaR with phosphoenolpyruvate. This limited overlap likely reflects our limited knowledge and the vastness of the search space, as each method explores distinct parts of this landscape. In scenarios where multiple candidate signals are predicted—such as with LeuO in this study—experimental validation is essential for accurately identifying the specific signal. Techniques like thermal shift assays and microscale thermophoresis are capable of detecting TF–metabolite interactions (Huynh and Partch, 2015; Jerabek-Willemsen et al, 2014); however, it is important to note that these interactions do not always correlate with functional effects, necessitating further characterization. Currently, the gold standard for functional validation are electromobility shift assays (Hellman and Fried, 2007), which, despite their reliability, are labor-intensive and impractical for large-scale applications. Consequently, we concentrated on one of our prediction cases, LeuO, and performed a comprehensive validation using two complementary methods. The functional validation of these interactions constitutes a significant bottleneck in achieving a fully mapped regulatory network. Given that inference methods like the one presented here can generate hypotheses for hundreds of TFs, there is a pressing need for a functional validation method that can match this throughput.

Collectively, these results demonstrate the potential of high-throughput approaches for the rapid discovery of TF–metabolite interactions, revealing numerous such interactions in *E. coli*. By facilitating the generation of hypotheses on TF input signals at the scale of hundreds of TFs, we anticipate that approaches like the one presented here will significantly accelerate advancements in the field of TF–metabolite interaction discovery.

# Methods

**Reagents and tools table**

| Reagent/resource | Reference or source | Identifier or catalog number |
| --- | --- | --- |
| **Oligonucleotides and other sequence-based reagents** | | |
| PCR primers | This study | Table EV8 |
| **Chemicals, enzymes, and other reagents** | | |
| Nitrocellulose filter | Durapore | HVLP04700 |
| Cobalt resin | Thermo | 89964 |
| Centrifugal filter | Millipore | UFC500324 |
| Protease inhibitor | Roche | 11836170001 |
| 5% TBE gel | Bio-rad | 4565015 |
| Thermal shift dye kit | Thermo | 4461146 |
| **Other** | | |
| Agilent QTOF 6550 Mass spectrometer | Agilent | |

## Bacterial strains and growth

Bacterial strains, media composition and growth conditions were chosen to match experimental procedures used to generate RNA-seq data in iModulonDB (Lamoureux et al, 2023) (Dataset EV2).

All samples were cultivated in biological triplicates in 14 ml tubes (Greiner, #187262) at 37 °C and 250 rpm agitation, in 3 ml of M9 minimal medium (42.2 mM $Na_2HPO_4$, 22 mM $KH_2PO_4$, 18.7 mM $NH_4Cl$, 8.5 mM $NaCl$, 1 mM $MgSO_4$, 0.1 mM $CaCl_2$, 20 µM $FeCl_3$, 2.4 µM $ZnCl_2$, 2.1 µM $CoCl_2$, 2.1 µM $Na_2MoO_4$, 1.3 µM $CuCl_2$, 2 µM $H_3BO_3$) supplemented with different carbon sources and supplements prior to growth experiments. For protein expression, strains from the ASKA library (Kitagawa et al, 2005) were grown in lysogeny broth medium containing 30 µg/ml chloramphenicol.

## Metabolites extraction

After overnight culture in M9 medium, cells were diluted to $OD_{600} = 0.05$ and cultivated until mid- to late-exponential phase (~2 to 5 h post inoculation, depending on conditions) to match the harvest $OD_{600}$ reported for RNA extraction (Lamoureux et al, 2023). Intracellular metabolites were then extracted by fast filtration. Briefly, a volume of culture corresponding to a 1 ml equivalent of $OD_{600} = 1$ was vacuum-filtered on a 0.45-µm nitrocellulose filter (Durapore, #HVLP04700). Immediately after filtration, metabolic processes in the filtered cells were quenched by putting the filter membrane into 1 ml of extraction solution (40% (v/v) acetonitrile, 40% methanol, 20% ddH$_2$O) precooled at −20 °C in six-well plates for lysis and metabolite extraction. Plates were sealed with parafilm, and metabolites were extracted overnight at −20 °C. After centrifugation of the extraction solution for 8 min at −2 °C and 21,000 × *g*, 650 µl of supernatant were collected and stored at −80 °C until measurement.

## Metabolomics

Extracts were measured in technical duplicates by double injection on an Agilent 6550 quadrupole time-of-flight mass spectrometer coupled to a Gerstel MPS2 autosampler using FIA-TOF-MS, as previously described (Fuhrer et al, 2011). In total, 5 µl of sample were injected into a constant 150 µl/min flow of running buffer (60:40 (v/v) isopropanol:water, 5 mM ammonium carbonate at pH 9, containing 3-amino-1-propanesulfonic acid and hexakis (1H, 1H, 3H-tetrafluoropropoxy)phosphazine for online mass axis correction). Mass spectra were recorded in negative ionization and high resolution modes with an acquisition rate of 1.4 spectra/s. Mass spectrometry data was merged, normalized and annotated in MATLAB R2021a. After spectral merging, counts were normalized by total ion count. Then, measured ions were annotated by mass-to-charge ratios to a reference list derived from a genome-wide reconstruction of *E. coli* metabolism within 0.001 Da mass tolerance. Biological and technical replicates were filtered for a coefficient of variation smaller than 15%.

## Estimation of transcription factors activities

RNA-seq data from the PRECISE 2.0 dataset and the *E. coli* transcriptional regulatory network from iModulonDB (Lamoureux et al, 2023), which was curated from RegulonDB v 10.5 (Santos-Zavaleta et al, 2019), were used to infer transcription factors activities (excluding sigma factors) using the decoupleR python package v 1.2.0 (Badia-i-Mompel et al, 2022) on TFs with at least three known target genes. To account for the incompleteness of the

regulatory network, the network was subsampled, as previously described (Ortmayr et al, 2019), for each TF within ten subnetworks containing 40 randomly chosen additional TFs for which activities were inferred. For each TF, the median of all subsampling simulations was calculated and used as the final activity value. Subsampling the regulatory network helps to avoid biased or unstable TF activity inference caused by uneven regulon sizes, missing data, or condition-specific interactions.

To assess the performance of different inference methods, conditions from iModulonDB (Lamoureux et al, 2023) that had single TF gene knockouts and matching wild-type strains were selected to compare inferred changes to expected directions; i.e., decreased TF activity in the corresponding TF mutant strain. Six methods from the decoupleR package were compared (Badia-i-Mompel et al, 2022) using a direction metric and median rank percentile metric (Ma and Brent, 2021). As a robust estimate, the median and interquartile range across all subsamples per TF in each condition was calculated. Accuracy was reported for each TF as the proportion of samples in which the activity is lower in the TF mutant compared to the wild-type strain (Fig. EV1A–C). Based on the accuracy of the direction metric, all TFs were filtered on baseline activity and minimum absolute activity difference, as thresholding on these criteria showed improved performance on correctly assigned TF activities (Fig. EV1D–G). To that aim, thresholds were determined as minimizing the $X^2$ $P$ value when dynamically thresholding across the range of corresponding values for increased correctly assigned TF activities. Specifically, TFs with a basal maximum activity lower than $-0.752$ (Fig. EV1D) or a maximum activity range lower than 0.166 (Fig. EV1E) were filtered out. The VIPER method was chosen to perform all further TF activity inference as it correctly assigned expected TF activity changes for the most conditions (Fig. EV1A–C).

## Determination of distances within the regulatory and metabolic networks

To obtain distances between TFs and metabolites, TF–gene interactions were obtained from the RegulonDB v10.5 transcriptional regulatory network (Santos-Zavaleta et al, 2019) and metabolic reactions from the iJO1366 E. coli metabolic model (Orth et al, 2011). All highly connected metabolites (involved in >50 reactions) and common cofactors were removed before calculation, and only intracellular metabolites and reactions were considered. For each TF–metabolite pair, a possible path between the two networks was searched if at least one of the target genes of the involved TF encodes an enzyme catalyzing a reaction present in the metabolic model. The shortest possible path was determined using the Dijkstra algorithm (Dijkstra, 2022), considering any possible direction. For a given TF–metabolite pair, if the metabolite is a product or a substrate of an enzyme whose gene is regulated by the involved TF, the assigned distance was zero. The distance was increased by one for each linked reaction needed to reach the closest regulated enzyme. If no path was found, the pair got assigned an infinite distance value. To determine the positions of metabolites within regulated subnetworks (i.e., downstream, upstream, or within), the directions of reaction were obtained from the iJO1366 E. coli metabolic model (Orth et al, 2011).

## Inference of TF–metabolite interactions

To predict TF–metabolite interaction candidates, we calculated pairwise Spearman correlations between TF activities and metabolite abundances. While a previous study has used Hill kinetics to predict TF–metabolite interactions (Lempp et al, 2019), Spearman correlation was chosen here because it relies on fewer assumptions and is tolerant to cases where the full dynamic range of the interaction is not captured. To decrease the effect of potential single-condition outliers and reduce the chances that some associations arise only because of a single data point, we used a bootstrapping approach and randomly subsampled 30 out of the 40 growth conditions over 1000 samples. For each sampling, Spearman's rank correlation coefficients were calculated between TF activities and metabolite abundances over the 30 conditions and used to generate a receiver-operator characteristics (ROC) curve using reported TF–metabolite interactions as true positives (Santos-Zavaleta et al, 2019) and all other pairs as negative cases. Correlation scores are reported for each TF–metabolite pair as the average correlation coefficient for that pair across all 1000 subsamples. Median ROC curves across all subsamples were used to set a correlation threshold controlling for a 0.1 false positive rate. For each TF–metabolite pair, a stability score was calculated as the proportion of subsamples above the correlation threshold.

To infer potential TF–metabolite interactions, pairs were filtered based on three criteria: their correlation, stability and distance scores. Based on the recovery of true positives, thresholds were determined for each metric (Fig. EV2) and pairs scoring above all thresholds were selected in the final lists of potential interactions (Dataset EV7). The correlation threshold was chosen to obtain a 0.1 false positive rate. The stability and distance thresholds of 0.128 and 0, respectively, were chosen as the values maximizing the proportion of retained true positives (Fig. EV2).

## Protein purification

Selected TFs were purified from overexpressing strains from the ASKA library (Kitagawa et al, 2005). Bacteria were grown in LB medium containing 30 µg/ml chloramphenicol. Growth was started from an overnight pre-culture into 100 ml of medium at an $OD_{600}$ of 0.05 at 37 °C under agitation (250 rpm). When the $OD_{600}$ reached 1, cultures were cooled to room temperature (RT) and 0.5 mM IPTG was added before resuming growth at RT under agitation for 16 h. Bacteria were then centrifuged at 3000 × g for 20 min at 4 °C. Pellets were resuspended in 10 ml of Lysis buffer (50 mM Tris-HCl, 500 mM NaCl, 10 mM imidazole, 5% glycerol, pH 8) containing freshly added protease inhibitors (Roche, #11836170001) and sonicated at 50% power for two cycles of 5 min using a Bandelin Sonopuls HD 2070 sonicator. Lysates were clarified by centrifugation at 3000 × g for 30 min at 4 °C. Supernatants were then incubated for 1 h on a rotating wheel with 500 µl of cobalt resin (Thermo, #89964) previously washed three times with Lysis buffer. Samples were then passed through empty gravity columns (Thermo, #29924) and washed once with Lysis buffer, twice with Lysis buffer containing 20 mM imidazole and twice with Lysis buffer containing 40 mM imidazole. After the final wash, columns were closed and 2 ml of Elution buffer (50 mM Tris-HCl, 500 mM NaCl, 200 mM imidazole, 5% glycerol, pH 8) were added. After incubating for 10 min, eluates were collected and then

buffer exchanged in storage buffer (50 mM Tris-HCl, 250 mM NaCl, 50 mM KCl, 10% glycerol, 0.1% Tween20, pH 8) using centrifugal filter units (Millipore, #UFC500324). Protein quality was assessed by SDS-PAGE, and concentration was measured using a Qubit fluorometer.

## Electromobility shift assays

Electromobility shift assays were performed as previously described (Trouillon et al, 2021). Briefly, Cy5-labeled DNA probes were generated by PCR from *E. coli* MG1655 genomic DNA using specific primers containing a Cy5 modification (Table EV1). Binding reactions were performed in EMSA buffer (10 mM Tris-HCl, 50 mM KCl, 5 mM $MgCl_2$, 5% glycerol, 0.1 mg/ml BSA, pH 7.5) in 20 µl final volume containing 0 or 10 mM of the tested metabolite, 10 nM DNA probe, 500 ng salmon sperm DNA and 2 µl of purified protein, added last. After 15 min of incubation at room temperature, 10 µl of sample was loaded on native 5% TBE gels (Bio-Rad, #4565015) and ran in 0.5× TBE buffer at 120 V at 4 °C for 1 h. Gels were visualized using a Fusion FX6 Edge imaging system, and band intensities were quantified using GelBox (Gulbulak et al, 2024).

## Thermal shift assay

Thermal shift assays were performed using the Protein Thermal Shift Dye Kit (Thermo, #4461146) following the manufacturer's instructions. Briefly, 2 µg of LeuO protein, 225 ng of *leuLABCD* promoter region DNA fragment and 0 or 10 mM of 2-isopropylmalate were incubated in 12.5 µl of EMSA buffer (containing no BSA) for 15 min at room temperature. Then 5 µl of Protein Thermal Shift Buffer and 2.5 µl of Protein Thermal Shift Dye were added, and melt curves were recorded on a QuantStudio 3 Real-Time PCR System (Applied Biosystems).

## Data availability

The datasets produced in this study are available in the following databases: Metabolomics data: MassIVE MSV000098001 (https://doi.org/10.25345/C5GQ6RF28).

The source data of this paper are collected in the following database record: biostudies:S-SCDT-10_1038-S44320-025-00132-2.

## Peer review information

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

## Acknowledgements

The authors are grateful to members of the lab of Prof. Bernhard O Palsson, especially Kevin Rychel, Ying Hefner, and Ye Gao, for their help and advice in using data from iModulonDB and reproducing growth conditions. We thank Prof. Mattia Zampieri for his valuable advice and discussions on transcription factor activity inference and the correlation analysis between transcription factor activities and metabolite concentrations. Funding was provided by the ETH Postdoctoral Fellowship program and an ETH Career Seed award (ETH Zürich) for JT.

## Author contributions

**Julian Trouillon**: Conceptualization; Investigation; Writing—original draft. **Alexandra E Huber**: Investigation. **Yannik Trabesinger**: Investigation. **Uwe Sauer**: Conceptualization; Writing—original draft.

Source data underlying figure panels in this paper may have individual authorship assigned. Where available, figure panel/source data authorship is listed in the following database record: biostudies:S-SCDT-10_1038-S44320-025-00132-2.

## Funding

## Disclosure and competing interests statement

The authors declare no competing interests. US is a member of the Advisory Editorial Board of Molecular Systems Biology. This has no bearing on the editorial consideration of this article for publication.

# Expanded View Figures

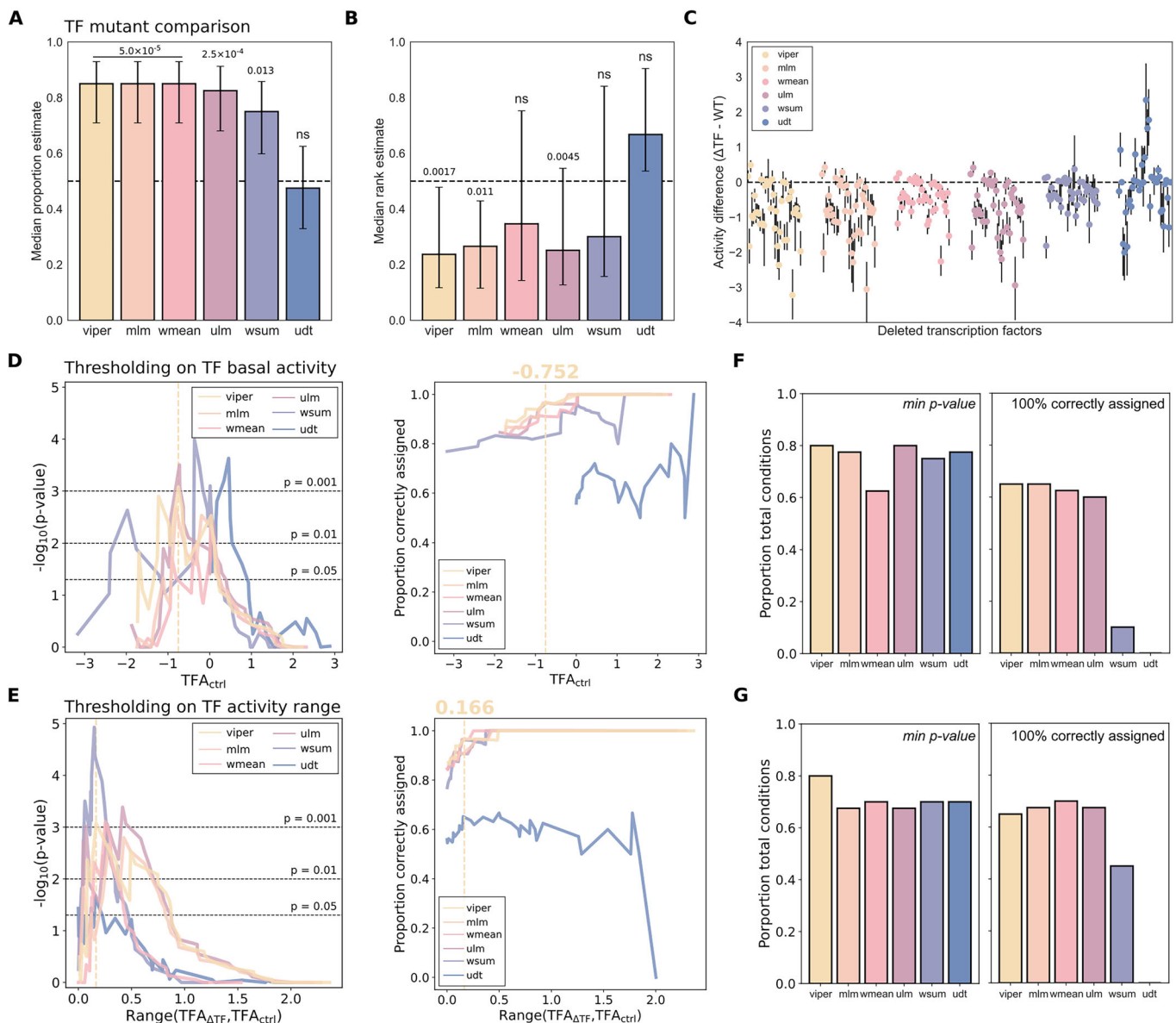

**Figure EV1.  Validation of transcription factor activity inference methods.**

(**A**) Proportion of TF mutant conditions with correct direction inference; i.e., the corresponding TF activity is lower in the TF knockout mutant sample compared to a control ($n = 40$). Bars indicate the 95% Wilson score intervals. (**B**) Median rank estimates of all deleted TFs across all TFs in TF mutant conditions ($n = 40$). Bars indicate interquantile ranges from TF activities across conditions. (**A, B**) *ns*: non-significant, Exact *P* values below 0.05 are indicated above the corresponding box from a two-sided binomial test against random distribution (dashed line) with Bonferroni correction. (**C**) TF activity difference (mutant against *corresponding wild-type*) for each TF mutant condition across the six tested methods ($n = 40$). (**D, E**) Left: Statistical significance from $X^2$ test for increased correctly assigned TF activities resulting from dynamical thresholding across ranges of values of (**D**) TF basal activity in the corresponding control condition or (**E**) TF activity range between the control and TF knockout mutant conditions. TFA: Transcription factor activity. Right: Proportion of correctly assigned TF activities from TF mutant conditions across a range of threshold values based on (**D**) TF basal activity in the corresponding control condition or (**E**) TF activity range between the control and TF mutant conditions. Threshold values minimizing the *P* value are shown as dotted vertical lines for the viper method when significant and are written in yellow above the corresponding dotted lines. (**F, G**) Proportion of experimental conditions kept after thresholding for (**F**) TF basal activity in the corresponding control condition or (**G**) TF activity range between the control and TF mutant conditions using either the threshold values minimizing the *P* value (left) or the threshold values that allow for 100% of correctly assigned TF activities when possible (right). (**A–G**) The six tested methods are multivariate linear model (mlm), weighted mean (wmean), virtual inference of protein-activity by enriched regulon analysis (viper), univariate linear model (ulm), weighted sum (wsum) and univariate decision tree (udt). All methods are colored following the same color scheme across all panels.

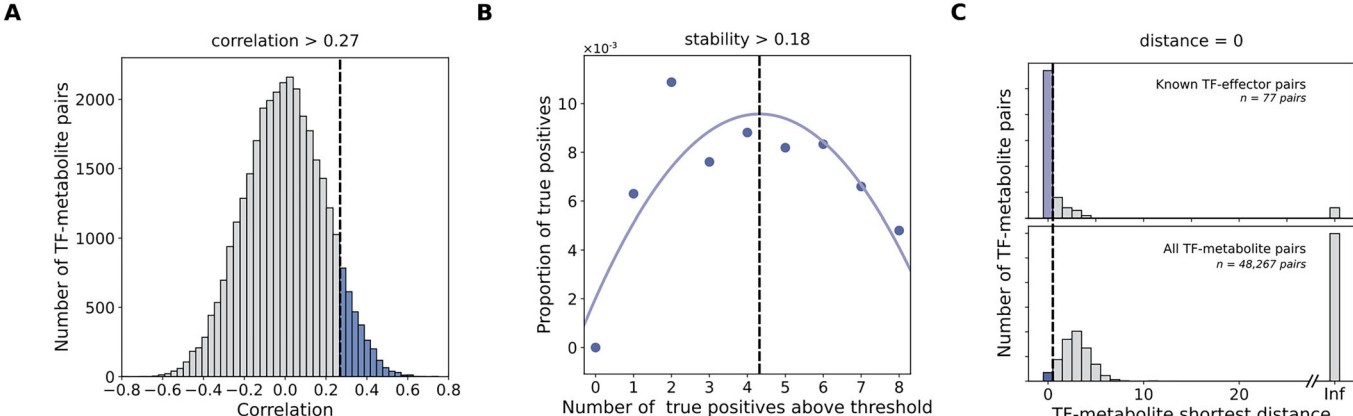

**Figure EV2. Filtering approach to identify candidate TF input signals.**

(A) Distribution of correlation scores across all TF–metabolite pairs. Bins above the False Positive Rate of 0.1 correlation threshold (dotted line) are colored in blue. (B) Proportion of known interactions recovered across a range of stability thresholds expressed as number of known interactions above threshold. The stability threshold was determined as the summit of a parabolic fit (dotted line). (C) Distribution of TF–metabolite distances for known interactions (top) or all pairs (bottom), with distance threshold of 0 shown as dotted line.

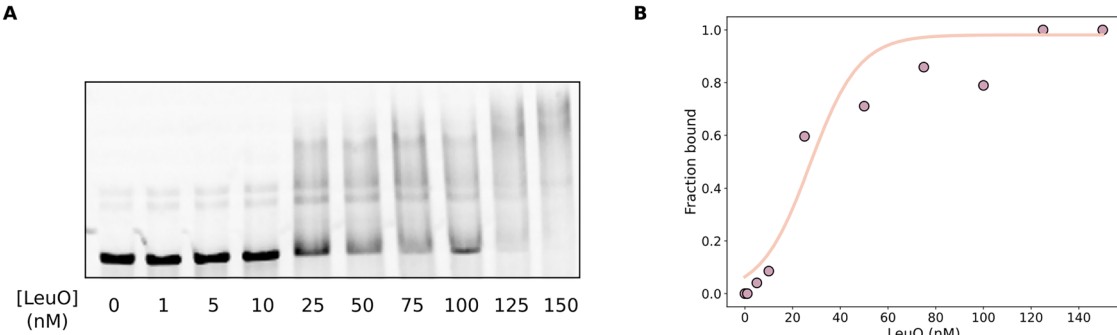

**Figure EV3. LeuO binding to the *leuLABCD* promoter region.**

(A) Gel shift experiment with increasing concentrations of LeuO protein and 0.5 nM of DNA fragment containing the *leuLABCD* promoter region. (B) Quantification of fractions of bound DNA from (A). The curved line represents a sigmoid fit of the data.

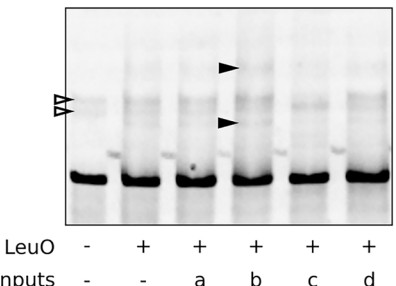

Candidate inputs:

a. pyruvate
b. 2-isopropylmalate
c. 3-isopropylmalate
d. 3-methyl-2-oxobutanoate

| LeuO | - | + | + | + | + | + |
| Inputs | - | - | a | b | c | d |

**Figure EV4.   Gel shift assay to identify LeuO signal molecule.**

A representative gel is shown out of four independent gel shift experiments with 0 (−) or 25 nM (+) of LeuO protein and 0.5 nM of DNA fragment containing the *leuLABCD* promoter region. 10 mM candidate inputs were added (+) to test their effect on LeuO binding. Hollow arrows indicate faint bands with unspecific shifts independent of LeuO. Black arrows indicate specific band shifts dependent of LeuO addition.

