## [Peer Review File · Molecular Systems Biology]

Predicting input signals of transcription factors in *Escherichia coli*

Julian Trouillon, Alexandra Huber, Yannik Trabesinger, and Uwe Sauer

Corresponding author(s): Uwe Sauer (sauer@imsb.biol.ethz.ch)

Review Timeline:

Submission Date:	24th Feb 25
Editorial Decision:	7th Apr 25
Revision Received:	3rd Jun 25
Editorial Decision:	26th Jun 25
Revision Received:	27th Jun 25
Accepted:	30th Jun 25

Editor: Jingyi Hou

Transaction Report:

7th Apr 2025

Manuscript Number: MSB-2025-12935

Title: Predicting input signals of transcription factors in Escherichia coli

Author: Uwe Sauer

Julian Trouillon

Alexandra Huber

Yannik Trapesinger

Dear Uwe,

Thank you for submitting your work to Molecular Systems Biology. We have now heard back from the three reviewers who agreed to evaluate your manuscript. As you will see from the reports below, the reviewers find the study potentially interesting and relevant. They raise, however, a series of concerns, which should be convincingly addressed in a major revision.

I think the reviewers' recommendations are relatively clear, so there is no need to reiterate the points listed below. All the issues raised by the reviewers need to be satisfactorily addressed. As you may already know, our editorial policy allows in principle a single round of major revision, and it is therefore essential to provide responses to the reviewers' comments that are as complete as possible. Please feel free to contact me in case you would like to discuss in further detail any of the issues raised by the reviewers.

On a more editorial level, we would ask you to address the following issues:

- Please provide a .docx formatted version of the manuscript text (including legends for main figures, EV figures and tables). Please make sure that the changes are highlighted to be clearly visible.
- Please provide individual production quality figure files as .eps, .tif, .jpg (one file per figure).
- Please provide a .docx formatted letter INCLUDING the reviewers' reports and your detailed point-by-point responses to their comments. As part of the EMBO Press transparent editorial process, the point-by-point response is part of the Review Process File (RPF), which will be published alongside your paper.
- Please note that all corresponding authors are required to supply an ORCID ID for their name upon submission of a revised manuscript.
- We replaced Supplementary Information with Expanded View (EV) Figures and Tables that are collapsible/expandable online (see examples in <http://msb.embopress.org/content/11/6/812>). A maximum of 5 EV Figures can be typeset. EV Figures should be cited as 'Figure EV1, Figure EV2' etc... in the text and their respective legends should be included in the main text after the legends of regular figures.

Additional Tables/Datasets should be labeled and referred to as Table EV1, Dataset EV1, etc. Legends have to be provided in a separate tab in case of .xls files. Alternatively, the legend can be supplied as a separate text file (README) and zipped together with the Table/Dataset file.

For the figures and tables that you do NOT wish to display as Expanded View figures, they should be bundled together with their legends in a single PDF file called *Appendix*, which should start with a short Table of Content. Each legend should be below the corresponding Figure/Table in the Appendix. Appendix figures and tables should be referred to in the main text as: "Appendix Figure S1, Appendix Figure S2, Appendix Table S1" etc. See detailed instructions regarding expanded view here: <https://www.embopress.org/page/journal/17444292/authorguide#expandedview>.

- Before submitting your revision, primary datasets (and computer code, where appropriate) produced in this study need to be deposited in an appropriate public database (see <http://msb.embopress.org/authorguide-dataavailability> <https://www.embopress.org/page/journal/17444292/authorguide#dataavailability>). Please remember to provide a reviewer password if the datasets are not yet public. The accession numbers and database should be listed in a formal "Data Availability" section (placed after Materials & Method) that follows the model below (see also <https://www.embopress.org/page/journal/17444292/authorguide#dataavailability>). Please note that the Data Availability Section is restricted to new primary data that are part of this study.

Data availability

- RNA-Seq data: Gene Expression Omnibus GSE46843 (<https://www.ncbi.nlm.nih.gov/geo/query/acc.cgi?acc=GSE46843>)

- [data type]: [name of the resource] [accession number/identifier/doi] ([URL or identifiers.org/DATABASE:ACCESSION])

-At EMBO Press we ask authors to provide source data for the main figures. Our source data coordinator will contact you to discuss which figure panels we would need source data for and will also provide you with helpful tips on how to upload and organize the files.

- Our journal encourages inclusion of *data citations in the reference list* to directly cite datasets that were re-used and obtained from public databases. Data citations in the article text are distinct from normal bibliographical citations and should directly link to the database records from which the data can be accessed. In the main text, data citations are formatted as follows: "Data ref: Smith et al, 2001". In the Reference list, data citations must be labeled with "[DATASET]". A data reference must provide the database name, accession number/identifiers and a resolvable link to the landing page from which the data can be accessed at the end of the reference. Further instructions are available at .

- We updated our journal's competing interests policy in January 2022 and request authors to consider both actual and perceived competing interests. Please review the policy <https://www.embopress.org/competing-interests> and update your competing interests if necessary. Please use the heading "Disclosure statement and competing interests".

- All Materials and Methods need to be described in the main text using our 'Structured Methods' format. According to this format, the Methods section includes a Reagents and Tools Table (listing key reagents, experimental models, software and relevant equipment and including their sources and relevant identifiers) followed by a Methods and Protocols section describing the methods, ideally using a step-by-step protocol format. The aim is to facilitate adoption of the methodologies across labs.

Please download and fill our Reagents and Tools Table template (.docx), which you can find in our author guidelines: <https://www.embopress.org/page/journal/17444292/authorguide#structuredmethods>.

An example of a Method paper with Structured Methods can be found here: <https://www.embopress.org/doi/10.15252/msb.20178071>.

-Regarding data quantification:

Please ensure to specify the name of the statistical test used to generate error bars and P values, the number (n) of independent experiments (please specify technical or biological replicates) underlying each data point and the test used to calculate p-values in each figure legend. Discussion of statistical methodology can be reported in the materials and methods section, but figure legends should contain a basic description of n, P and the test applied.

Graphs must include a description of the bars and the error bars (s.d., s.e.m.).

- Please provide a "standfirst text" summarizing the study in one or two sentences (approximately 250 characters, including space), three to four "bullet points" highlighting the main findings and a "synopsis image" (550px width and 400-600 px height, PNG format) to highlight the paper on our homepage.

Here are a couple of examples:

<https://www.embopress.org/doi/10.15252/msb.20199356>

<https://www.embopress.org/doi/10.15252/msb.20209475>

<https://www.embopress.org/doi/10.15252/msb.209495>

When you resubmit your manuscript, please download our CHECKLIST (<https://www.embopress.org/pb-assets/embo-site/EMBO%20Press%20Author%20Checklist-1642513524327.xlsx>) and include the completed form in your submission.

Please note that the Author Checklist will be published alongside the paper as part of the transparent process (<https://www.embopress.org/page/journal/17444292/authorguide#transparentprocess>).

If you feel you can satisfactorily deal with these points and those listed by the referees, you may wish to submit a revised version of your manuscript. Please attach a covering letter giving details of the way in which you have handled each of the points raised by the referees. A revised manuscript will be once again subject to review and you probably understand that we can give you no guarantee at this stage that the eventual outcome will be favorable.

I look forward to receiving your revised manuscript soon.

Kind regards,
Jingyi

We realize that it is difficult to revise to a specific deadline. In the interest of protecting the conceptual advance provided by the work, we recommend a revision within 3 months (6th Jul 2025). Please discuss the revision progress ahead of this time with the editor if you require more time to complete the revisions. Use the link below to submit your revision:

IMPORTANT: When you send your revision, we will require the following items:

1. the manuscript text in LaTeX, RTF or MS Word format
2. a letter with a detailed description of the changes made in response to the referees. Please specify clearly the exact places in the text (pages and paragraphs) where each change has been made in response to each specific comment given
3. three to four 'bullet points' highlighting the main findings of your study
4. a short 'blurb' text summarizing in two sentences the study (max. 250 characters)
5. a 'thumbnail image' (550px width and max 400px height, Illustrator, PowerPoint or jpeg format), which can be used as 'visual title' for the synopsis section of your paper.
6. Please include an author contributions statement after the Acknowledgements section (see <https://www.embopress.org/page/journal/17444292/authorguide>)
7. Please complete the CHECKLIST available at (<https://bit.ly/EMBOPressAuthorChecklist>). Please note that the Author Checklist will be published alongside the paper as part of the transparent process (<https://www.embopress.org/page/journal/17444292/authorguide#transparentprocess>).
8. When assembling figures, please refer to our figure preparation guideline in order to ensure proper formatting and readability in print as well as on screen:
<https://bit.ly/EMBOPressFigurePreparationGuideline>
See also figure legend guidelines: <https://www.embopress.org/page/journal/17444292/authorguide#figureformat>
9. Please note that corresponding authors are required to supply an ORCID ID for their name upon submission of a revised manuscript (EMBO Press signed a joint statement to encourage ORCID adoption). (<https://www.embopress.org/page/journal/17444292/authorguide#editorialprocess>)
Currently, our records indicate that the ORCID for your account is 0000-0002-5923-0770.

Link Not Available

11. Include a Reagents and Tools Table as part of the Methods section, which can be downloaded from our author guidelines (<https://www.embopress.org/page/journal/17444292/authorguide#structuredmethods>)

*** PLEASE NOTE *** As part of the EMBO Press transparent editorial process initiative (see our Editorial at <https://dx.doi.org/10.1038/msb.2010.72>), Molecular Systems Biology publishes online a Review Process File with each accepted manuscripts. This file will be published in conjunction with your paper and will include the anonymous referee reports, your point-by-point response and all pertinent correspondence relating to the manuscript. If you do NOT want this File to be published, please inform the editorial office at msb@embo.org within 14 days upon receipt of the present letter.

Reviewer #1:

Predicting input signals of transcription factors in 1 Escherichia coli

The authors developed a systematic workflow for identifying transcription factor (TF) input signals in E. coli, which in principle

can be applied to other bacteria. This approach is an expansion of a previously published approach based metabolomics and transcriptomics data and correlate TF activities with metabolite abundances to predict allosteric metabolites that bind to specific TFs. TF activity is inferred based on the expression levels of the genes comprising the regulon of a given TF. The methodological strategy is essentially that published previously (PMID: 31578326) applied only to a single carbon growth condition change. In this paper the authors analyzed transcriptomic data from the publicly available PRECISE2.0 dataset, and identified 40 different growth conditions for which they obtained high-throughput metabolomics data. This expansion enabled the prediction of a larger number of allosteric-TF pairs. This approach requires previous knowledge of a set of regulated genes by each TF candidate, which was obtained from the RegulonDB database.

First, the authors evaluated six different computational methods to determine their ability to predict the expected decrease in TF activity in mutant strains from the PRECISE2.0 dataset, where specific TFs had been knocked out. They found that VIPER performed best and subsequently used it with a selected subset of 40 datasets from PRECISE2.0. The diversity of this subset (representing approximately 10% of the full dataset) was assessed, revealing that it retained an average of 72% of the maximum range of TF activity values observed across the entire dataset for the 173 measured TFs. The growth conditions corresponding to these 40 datasets were replicated and the corresponding metabolomics data was generated.

The analysis integrating transcriptomic and metabolomic data yielded an average correlation across 48,267 TF-metabolite pairs for the 173 TFs. These pairs were filtered using multiple criteria: a false positive rate threshold of 0.1, a stability threshold defined as the value that maximized the retention of known TF-metabolite interactions, and a distance metric representing the number of consecutive reactions in the metabolic network required to connect the metabolite to the nearest enzyme directly regulated by the TF. This filtering process resulted in a final list of 103 TF-metabolite pairs corresponding to 42 different TFs. The variation in the abundance of the metabolite was significant with a 3.34 fold change. One metabolite as signal at least for 42 TFs were predicted. For CRP, 23 metabolites including cAMP were predicted as signal which fits with the fact that it is a global TF.

Based on prior analyses, the authors experimentally evaluated three candidate metabolites and identified 2-isopropylmalate as the signaling metabolite by measuring its effect on LeuO's DNA-binding activity. This is the only case that the authors experimentally tested to validate their predictions. Finally, the authors discuss the multiple limitations of this approach in order to be more widely applied.

There is no doubt, as the authors of this and similar approaches claim, that, first, these interactions are an essential component to characterize and physiologically interpret the regulatory network governing gene expression, and second, there is a clear lack of well-established high throughput methodologies in order to eventually characterize the regulatory network for a given bacteria. Therefore, this work is certainly relevant not only for the community working in E.coli, but in principle also in microbial biology, and eventually in other domains of life.

The strategy to predict allosteric physiologically significant interactions with TFs is based on a clever combination of several components of biological knowledge and large amounts of data. Therefore, it is no surprise, as the authors mention in the discussion, that the strategy as such has important limitations for its wider application, not only in E.coli where a large body of knowledge is available, but in the large number of other sequenced microorganisms where knowledge of the regulatory network is much more limited.

At the core of the method that links potential allosteric metabolites with their corresponding TFs, is the identification of a correlation of the metabolite abundance with the TF activity. On the side of the metabolites, problem one is that multiple metabolites show similar abundance dynamics, and second, the MS-1 method cannot always distinguish among metabolites. In the Lempp et al paper, one of the criteria was to select those metabolites that show a Hill-curve behavior with TFs. This criteria is missing in this manuscript, and it is not even mentioned. This is an important aspect to be addressed in this manuscript.

On the side of identifying the activity of TFs, as already mentioned the authors evaluated six different computational methods, identifying VIPER as the most effective one. Incidentally, there is no clear definition, explanation of what the authors mean by "TF activity", and how or whether it differs from high TF concentration. This should be clarified at the beginning of the paper (line 54 or so).

A really relevant major concern is the availability of the data generated in this work, such as the metabolomic profiles, and additional processing of valuable elements supporting their results. For instance, the authors should make available the list of alternative candidate metabolites for each TF with the corresponding information used in their evaluation, including, among others, the distance to the closest enzyme, which enzyme is it, and additional relevant criteria in their selection.

Also, there is no mention at all on the challenge of assigning TF activity in the frequent cases of target genes subject to regulation by multiple TFs and under different growth conditions.

No mention to other publications. As mentioned, this work follows the Lempp approach, which is far from being the only one that has been published to widely identify allosteric metabolites. However, the authors made no mention of alternative approaches. Here is a short list of some alternative strategies: PMID: 31665428 and the Allosteric Database (ASD), 29757429, PMID: 31871208, PMID: 22122470, 16818878, and this is not an exhaustive list.

Q: Have the authors evaluated how different are the results if the biochemical distance of a candidate allosteric metabolite is not required to be directional? Or how sensitive is this direction to changes on growth conditions? Certainly, if direction is not so relevant, it would make this methodology more widely applicable. This should be mentioned, particularly given that the authors are somehow selling the relevance not only of the specific novel results, but of the overall strategy as well.

Given the rich set of data and processing, it is a pity that the authors do not enrich this work by analyzing in more detail their results. For instance, they could search for correlations and report if found or not between the range of the TF activity, the range of regulated gene changes and if that affects their predictability; frequently the so-called dual regulators might be activators with only negative autoregulation. Does autoregulation affect TF activity? Not all known metabolites were found, like the case of BetI

for which choline is the known effector or for CysB whose effector is known (O-acetyl-L-serine (OAS)), as well as for AraC, MetR, FucR, among others with known effector. Any insight about those that failed?

The value of publications is not limited to their data, but also to their reproducibility, as well as to their ability to generate knowledge in a wider context and convey such knowledge to larger audiences, particularly in journals committed to large communities such as this one. I would like the authors to mention and even show the lack or amount of consistency between this final "TF activity" coming out of such computations where the known regulons are required, versus the direct quantification of differential mRNA TF levels from the transcriptomic datasets. The average correlation between mRNA levels and the corresponding proteins in bacteria is quite high as shown by the work of Terry Hwa. This might also offer the opportunity to utilize the distinction between TFs that bind in the conformation with the allosteric metabolite bound and those that bind in their absence.

I strongly suggest that authors devote some thinking and write down a paragraph explaining what is different in VIPER for it to perform better than the other methods. Incidentally, in all graphs showing the comparative results of the different methods (Fig 1S), it is the VIPER lines that should have a clearly distinguishable color different from the other methods. Their current graph gives visual priority to the udt method in blue.

It seems that the growth conditions of the transcriptomic analyses with those of metabolomics are not exactly the same or had some variations in the culture media. Please clarify.

In conclusion before minor details:

The problem being addressed is no doubt highly relevant. The strategy is not novel, but this work is a significant advance in the field given the comprehensiveness of its implementation. The paper is well written, with complex massive results well presented. However, as they recognize in the discussion, the limitation of the current implementation is the need for a large number of constraints to be satisfied: a combined large dataset of transcriptomics and metabolomics under the same growth conditions, and for a large number of growth conditions; transcriptomics should include KO of TFs for validation of the directional change in TF activity. Furthermore, previous knowledge of the set of target genes or regulons for the TFs whose allosteric metabolites are to be discovered is needed. Even if all these requirements are met, as in E.coli as here implemented, there is still the inherent difficulty of the fact that several metabolites follow the same dynamical changes with TF activity. This will have no apparent solution in the near future as this is inherent in the biology of bacteria. (Q: Have you assessed whether precisely given these "dancing together" the regulatory systems should use only one metabolite as a signal of such metabolic cellular transition? How similar is the chemical structure of such co-dancing metabolites?). Finally, there is the technical current limitation of mass spec metabolites cannot be distinguished.

Additional specific concerns and questions:

In line 76 the authors refer to Figure 1S as evidence to show that 34 out of 40 pairs were correctly assigned. But Fig 1S has 12 different panels, so it will be difficult for readers to identify where in this very rich figure to find the precise support for this statement. See also Fig 1 panel d with no distinctive description within the figure. Each panel in the figures should have a unique title to facilitate the understanding of the figures without the need to get deep into the footnote text.

Line 300. Since the metabolomics analysis was performed on the extracellular medium without lysing the cells, is it possible that intracellular metabolites remain undetected, meaning that only external signals are being captured?

Minor observations.

Line 321 "the E. coli 320 transcriptional regulatory network from RegulonDB v 10.5 curated from iModulonDB (Santos-Zavaleta et al, 2019)". This is a confusing sentence.

The authors in the introduction make a good coverage of relevant citations, which nonetheless covers recent years, but very few to recognize for instance the very first characterization of allosterism in TFs.

Line 165-166: The sentence quoting ref (Santos-Zavaleta et al, 2019) is wrong.

There was a typo in a footnote of a supplementary material, Figure S1: "..(h,i) Porportion of experimental conditions.."

Reviewer #2:

In this work, the authors utilize a compendium of transcriptomics data (PRECISE 2.0) to correlate TRN signals (computed with VIPER + regulonDB) with metabolite levels across 40 conditions in order to hypothesize and validate new TF-metabolite interactions in E. coli. This work is next in a line of exciting endeavors to fill a major gap in our understanding of model organisms - our limited knowledge of small metabolite regulators of protein function, which until recently has been pieced together painstakingly through hypothesis-driven science. New methods are making it possible to search for these interactions en masse, which should have a transformative effect on systems biology both for data analysis and modeling. The approach of inferring TF activities from gene expression data has seen a resurgence in recent years and has shown itself to be quite successful in microbes; the missing piece has been metabolomics data which is still fairly limited in terms of publicly available

datasets. As the authors fill this critical gap with metabolomics data for 279 metabolites over 40 conditions and experimentally validate their predictions, I applaud the effort. My primary concerns with the work are that I have some doubt about many of the predicted associations from their workflow (which could be alleviated through further analysis or experimental validation), and the presentation in terms of Figures, description of Results, and Methods could probably be improved. Specific comments are below.

Major Comments:

- The authors utilize a Spearman correlation between TF activities and metabolite abundances to assign putative regulatory roles. Would it make sense to account for the expression change of the regulator itself as well? For example, if a TF activity was constant, but the TF expression was down (due to some other regulator), and a metabolite level was up, that metabolite actually be an activator despite no direct association between activity and metabolite levels. Did the authors look at the role of TF expression itself on their predicted associations?
- Is there any evidence in the data of multiple metabolites regulating TFs? PurR is thought to be regulated by two transcription factors - I'm curious whether their approach based on Spearman correlations (i.e. assuming a single regulatory factor) could identify this or not. Related to this point, I understand that PurR was not identified as a statistical association - I'm also curious if for these 'failed' cases, whether any signal can be seen in the data that may have been below the top thresholds applied in the final predicted associations.
- This is a matter of opinion but Figures 2-4 seem a bit drab and uninspiring, with key results somewhat difficult to glean. Inserting a few case studies e.g. on successful inference of known TF-metabolite interactions where we can see what the raw data looks like and how strongly the identified associations appear, would be helpful. Similarly, Fig 6 validates their predicted association but does not show the data for that association in the first place i.e. the expression of the LeuO regulon vs the metabolite level of the hypothesized effector. Figure 5 is the only figure that felt informative and interesting to read.
- It would be nice to compare their results to other high throughput experimental metabolite-protein interaction measurement methods (e.g. the recent Cell paper "Ligand interaction landscape of transcription factors and essential enzymes in *E. coli*" as well as the senior authors work such as <https://doi.org/10.1016/j.cell.2017.12.006>). Were the interactions that were inferred in this paper also detected in those works? Just trying to get a sense for the comparative discovery value of these approaches, and wondering if these studies could be a source of high-throughput validation of the predictions made in this work.
- On the choice of conditions for metabolomics data collection - Were these 40 conditions selected based on the 47 TF-associated conditions mentioned earlier? How much did the conditions seem to impact which TFs had a metabolite association (related to Figure 2a)? Do TFs activated and not activated make sense given the conditions chosen?
- Around line 128 where prediction ability is discussed - can the authors plainly state what number of TFs they were able to successfully identify a known metabolite interaction from this approach (I believe the number is 5)? I understand thresholds apply but they can select some threshold and report Precision and Recall both if they like. At line 144 they state: "For positively correlating pairs, applying the stability and distance filters increased the proportion of recovered known interactions among predicted pairs about 100-fold, resulting in a final list of 103 TF-metabolite pairs, only 5 of which were previously known (Figure 5a, Table S7). Overall, this approach identified candidate input signals for 42 TFs." I must misunderstand something, because they say they increased the proportion of known associations 100-fold, and ended up with 5 known associations. Could the authors please clarify these statistics?
- I see in Figure 5a the known associations - e.g. Crp-cAMP that the authors discuss in the text. The authors comment on why these believe these associations appeared in their approach around line 148, but I don't understand this reduction to only 8 cases. They produced predicted associations for 42 TFs. They then reduce this to 18 TFs that had previously known small molecule associations. From these 18, they then seem to look only at 8 metabolites that are activators and measured in their study, if I understand correctly, and state they correctly inferred 5 of those, while the others known associated metabolites are nearby the predicted ones in the metabolic network. If I understand these filters correctly, why were only activating metabolites considered?
- Related to the above comment - could the authors comment on why other well-known associations may not have been picked up? i.e. Cra-FDP, pyruvate with PdhR/IcIR, ArgR-arginine and so on?
- What is the relevance of the metabolic network distance (Figure 4e)? They seem to use it as a criteria for selection of TF-metabolite interactions but I don't understand the principle. "more permissive choice to increase chances of discovering interactions involving metabolites also outside of the regulated enzymatic reactions." It seems like they did not want to include metabolites that could be affected indirectly through regulon perturbation (i.e. the TF affects the enzyme, which affects the metabolite, as opposed to the metabolite affecting the TF - but cases like arginine regulating ArgR would go against this strategy). I understand the challenge of disentangling cause and effect, I'm just not clear if applying a distance threshold is universally helpful. Could the authors comment on this?
- Despite the care taken by the authors to optimize their workflow, it unclear how credible the predictions in Figure 5a are intended to be. Do we expect the SoxS is regulated by CTP/UTP and citrate/isocitrate? That Fnr is regulated by Threonine? These don't have the clear functional ties that their LeuO case has, at first glance at least. I'm sure that there is some kind of functional connection underlying many of these associations, but given the complexities of metabolic cause and effect - how do the authors propose we narrow in on the most likely associations to validate? It seems like LeuO was chosen as its predicted effectors had an apparent topological logic to their putative regulatory role. The authors mention topologies in their discussion of Figure 5, but could the details here be presented on a case-by-case basis in Figure 5a as opposed to simple lump statistics in Figure 5b? Maybe this could be part of Supplementary Table 7 as well.
- The glutamine-gadX prediction is probably important enough to warrant experimental validation. A second validation case would go a long way as well toward addressing potential skepticism around many of the predicted interactions.

Minor Comments:

- Should the Introduction mention sigma factors as well or are these included in transcription factors? Not clear whether these were included, as their known regulators are not necessarily small molecules I believe, but they nevertheless greatly impact gene expression in bacteria.
- The Biorxiv manuscript for the PRECISE2.0 database is now published - it may be appropriate to update the link.
- Line 78 in the Results: "The PRECISE2.0 dataset includes 47 conditions related to 23 TFs," - Unclear what 'related' means or how these 47 conditions were selected from the entire dataset. Was this based on some activity threshold for a set of regulons that was examined?
- Line 79 in the Results: "In 83% of these cases," Some discussion of the minority of cases that did not work would be warranted, as well as how the 'expected' direction of perturbation was identified. Was this straightforward or were there any ambiguous cases? This paragraph is described in such a general way that the results seem a bit vague and less convincing than necessary (despite the strong signal seen in the figure).
- Line 121 - How necessary was this bootstrapping approach? What kinds of problematic issues arose when this step was originally excluded? Per a more technical question I had further below, I had a hard time understanding the rationale here and the problem that this approach solves. What is a "poorly matched growth condition" for example?
- Line 223 - 41 TFs are mentioned here, but 42 were mentioned earlier in line 147 (later 41+Crp were mentioned, so maybe 42 is the correct number?).
- Could Figure 2a be somehow organized to impart more information? TFs could be sorted by regulon size, organized by functional category or at least sorted by range in activity.
- Fig 3A - PCA plots typically show the explained variance along the X and Y axis. A biplot would be even more informative, to show the top metabolites driving separation in the top 2 PCs.
- Since VIPER was chosen as the TF activity inference method based on performance evaluated in Figure S1 - some brief introduction into how it works would be warranted.
- Line 323 of the Methods is not clearly written to me: "To account for incompleteness of the regulatory network, the network was subsampled, as previously described (Ortmayr et al, 2019), for each TF within ten subnetworks containing 40 randomly chosen additional TFs for which activities were inferred. For each TF, the median of all subsampling simulations was calculated and used as final activity value." I'm having a hard time understanding what was done here, why the network needs to be subsampled, or what it means for "the median of all subsampling simulations" to be calculated (presumably this means the median of the decoupleR-computed activities across different subsampled regulons?).
- Similarly, line 335 of the Methods was also confusing to me: "Based on accuracy on the direction metric, all TFs were filtered on baseline activity and minimum absolute activity difference as thresholding on these criteria showed improved performance on correctly assigned TF activities. To that aim, thresholds were chosen as minimizing the X2 p-value when dynamically thresholding across the range of corresponding values for increased correctly assigned TF activities." Could this description be expanded and stated in a simpler way to make it easier to understand what was done? Multiple thresholds/filters were discussed but it's not clear what values were chosen or why/how.

Reviewer #3:

This study by Sauer and colleagues uses available transcriptomic datasets with high-throughput metabolomics to identify transcription factor-metabolite interactions that could impinge on transcription factor activity. Through an intricate analyses, the study identifies 76 novel metabolite-TF interactions and the data is validated by examining the predicted input signal for LeuO. Overall, this exhaustive study provides a useful resource to investigate various metabolism-dependent gene-regulatory networks in *E. coli* and the approach could very well be used to study similar networks in other bacteria.

However, the following concerns need to be addressed to further strengthen this study:

Line 77 to 81: The expression of genes corresponding to targeted TFs were checked in the presence of specific metabolites. For this analysis, the techniques and conditions used to quantify gene expression is not clear. It would be useful if the parameters are mentioned in the methods.

Line 80: In figure 1D, the authors have shown the change TF activity when a metabolite is added in the growth medium, compared to a paired control condition. However, Figure1D legend says "activity difference between the TF knockout mutant and the corresponding wild-type control conditions". It is difficult to interpret the data from the text provided in result section vs corresponding figure legend. Authors should provide enough information or rewrite the result section and figure legends for clear understanding.

Line 107 and 108: "The metabolites were extracted at the exponential phase, where, the cells were harvested at OD600 reported for RNA extraction" (as per methods, line 298-299). However, considering the diversity in conditions and strains used in this study, the growth in these conditions could vary across samples. Further, this will eventually lead to variation in time required to attain exponential phase and OD at exponential phase. It is not clear how the authors manage to identify the correct timepoint for extraction, especially when both of above factors vary largely across the samples.

The difference in metabolite quantification could also be contributed by the difference in extraction efficiency. Was this taken into consideration? If so, a line on how this was done will be useful.

Line 189-190: Out of 4 candidate inputs for LeuO effect of only one metabolite is validated. For other three candidates, it is possible that derivatives of, or chemical modifications (acetylation, lactylation etc.) caused by, input metabolites could influence TF activity. Is it possible to identify such cases from the existing data?

Figure S4: The gel shift assay shows a very faint band on addition of 2-isopropylmalate and is difficult to interpret.

It is not an absolute requirement but an in-vivo validation of TF-metabolite interaction would provide more impact to the work.

Line 115-116: Figure reference is missing.

Line 125: Shouldn't Fig. 4B be Fig. 4C?

Reference missing for line 298-299.

Supplementary figure: S1H and S1I: x-axis is not labelled.

Reviewer #1:

The authors developed a systematic workflow for identifying transcription factor (TF) input signals in *E. coli*, which in principle can be applied to other bacteria. This approach is an expansion of a previously published approach based metabolomics and transcriptomics data and correlate TF activities with metabolite abundances to predict allosteric metabolites that bind to specific TFs. TF activity is inferred based on the expression levels of the genes comprising the regulon of a given TF. The methodological strategy is essentially that published previously (PMID: 31578326) applied only to a single carbon growth condition change. In this paper the authors analyzed transcriptomic data from the publicly available PRECISE2.0 dataset, and identified 40 different growth conditions for which they obtained high-throughput metabolomics data. This expansion enabled the prediction of a larger number of allosteric-TF pairs. This approach requires previous knowledge of a set of regulated genes by each TF candidate, which was obtained from the RegulonDB database.

First, the authors evaluated six different computational methods to determine their ability to predict the expected decrease in TF activity in mutant strains from the PRECISE2.0 dataset, where specific TFs had been knocked out. They found that VIPER performed best and subsequently used it with a selected subset of 40 datasets from PRECISE2.0. The diversity of this subset (representing approximately 10% of the full dataset) was assessed, revealing that it retained an average of 72% of the maximum range of TF activity values observed across the entire dataset for the 173 measured TFs. The growth conditions corresponding to these 40 datasets were replicated and the corresponding metabolomics data was generated.

The analysis integrating transcriptomic and metabolomic data yielded an average correlation across 48,267 TF-metabolite pairs for the 173 TFs. These pairs were filtered using multiple criteria: a false positive rate threshold of 0.1, a stability threshold defined as the value that maximized the retention of known TF-metabolite interactions, and a distance metric representing the number of consecutive reactions in the metabolic network required to connect the metabolite to the nearest enzyme directly regulated by the TF. This filtering process resulted in a final list of 103 TF-metabolite pairs corresponding to 42 different TFs.

The variation in the abundance of the metabolite was significant with a 3.34 fold change. One metabolite as signal at least for 42 TFs were predicted. For CRP, 23 metabolites including cAMP were predicted as signal which fits with the fact that it is a global TF.

Based on prior analyses, the authors experimentally evaluated three candidate metabolites and identified 2-isopropylmalate as the signaling metabolite by measuring its effect on LeuO's DNA-binding activity. This is the only case that the authors experimentally tested to validate their predictions. Finally, the authors discuss the multiple limitations of this approach in order to be more widely applied.

There is no doubt, as the authors of this and similar approaches claim, that, first, these interactions are an essential component to characterize and physiologically interpret the regulatory network governing gene expression, and second, there is a clear lack of well-established high throughput methodologies in order to eventually characterize the regulatory network for a given bacteria. Therefore, this work is certainly relevant not only for the community working in *E. coli*, but in principle also in microbial biology, and eventually in other domains of life.

The strategy to predict allosteric physiologically significant interactions with TFs is based on a clever combination of several components of biological knowledge and large amounts of data. Therefore, it is no surprise, as the authors mention in the discussion, that the strategy as such has important limitations for its wider application, not only in *E. coli* where a large body of knowledge is available, but in the large number of other sequenced microorganisms where knowledge of the regulatory network is much more limited.

We thank reviewer 1 for the detailed analysis and positive comments.

At the core of the method that links potential allosteric metabolites with their corresponding TFs, is the identification of a correlation of the metabolite abundance with the TF activity. On the side of the metabolites, problem one is that multiple metabolites show similar abundance dynamics, and second, the

MS-1 method cannot always distinguish among metabolites. In the Lempp et al paper, one of the criteria was to select those metabolites that show a Hill-curve behavior with TFs. This criteria is missing in this manuscript, and it is not even mentioned. This is an important aspect to be addressed in this manuscript.

The choice of using Hill kinetics to identify metabolite-TF pairs does not help with distinguishing metabolites that show the same abundance dynamics or for metabolites that have the same masses using MS1 methods. The way Lempp et al mostly dealt with metabolites of similar abundance dynamics was through additional filtering steps, such as with the distance criterion. Indeed, the Lempp et al paper used a different correlation strategy where the TF-metabolite pairs were first correlated linearly and then a non-linear fit was used based on Hill kinetics. This approach relies on several assumptions, including that the maximal TF activity stays constant over time and the definition of a constrained Hill coefficient. Here, we decided to use Spearman correlation because it relies on less assumptions, is easier to implement for non-expert labs, and is more tolerant of cases where the full dynamic range of the interaction is not captured.

We now comment on this aspect in the corresponding method paragraph.

On the side of identifying the activity of TFs, as already mentioned the authors evaluated six different computational methods, identifying VIPER as the most effective one. Incidentally, there is no clear definition, explanation of what the authors mean by "TF activity", and how or whether it differs from high TF concentration. This should be clarified at the beginning of the paper (line 54 or so).

We defined the activity of a TF as the functional influence that a TF exerts on the expression of its direct target genes (its regulon), inferred from the collective expression pattern of those targets. The concentration of a TF is expected to affect this measure; however, a TF's mRNA levels are known to be poor proxy of its activity (PMID: 30462289). For more details, see answer on TF mRNA levels below. We clarified the definition of activity in the corresponding section.

A really relevant major concern is the availability of the data generated in this work, such as the metabolomic profiles, and additional processing of valuable elements supporting their results. For instance, the authors should make available the list of alternative candidate metabolites for each TF with the corresponding information used in their evaluation, including, among others, the distance to the closest enzyme, which enzyme is it, and additional relevant criteria in their selection.

We agree and have made several adjustments to improve on this point. The raw metabolomics data were deposited in the MassIVE database (Accession ID MSV000098001) and the processed metabolomics profiles are available as Supplementary Table 6. We also made available all source data for all figures as part of the Source Data files. Additionally, we modified the Supplementary Table 7 containing paired information and added suggested key information.

Also, there is no mention at all on the challenge of assigning TF activity in the frequent cases of target genes subject to regulation by multiple TFs and under different growth conditions.

Indeed, many genes are regulated by multiple TFs, which does pose a challenge for inferring TF activity from transcriptomic data, which is considered by the VIPER method that we used. Specifically, TF activity is inferred from collective behavior of the entire regulon, which helps dilute the confounding influence of a few targets being co-regulated by other TFs. While this doesn't completely eliminate the issue, it mitigates it by emphasizing coherent, coordinated signal across the entire regulon. To further ensure proper inference, we only used TFs which have at least 3 known targets, as inferring activities from less target genes would be prone to inaccuracies for the above reasons. Our tests on the different methods using data from TF mutants show that our method using VIPER can correctly capture changes in the activity of individual TFs in the vast majority of cases.

We have added a clarifying sentence as part of the newly added paragraph discussing TF inference and the VIPER method in the Discussion.

No mention to other publications. As mentioned, this work follows the Lempp approach, which is far from being the only one that has been published to widely identify allosteric metabolites. However, the authors made no mention of alternative approaches. Here is a short list of some alternative strategies: PMID: 31665428 and the Allosteric Database (ASD), 29757429, PMID: 31871208, PMID: 22122470, 16818878, and this is not an exhaustive list.

Beyond the Lempp approach as the most closely related to this work, we had also referred to multiple other relevant papers. We now have also included several of those suggested by the reviewer in the Introduction and Discussion.

Q: Have the authors evaluated how different are the results if the biochemical distance of a candidate allosteric metabolite is not required to be directional? Or how sensitive is this direction to changes on growth conditions? Certainly, if direction is not so relevant, it would make this methodology more widely applicable. This should be mentioned, particularly given that the authors are somehow selling the relevance not only of the specific novel results, but of the overall strategy as well.

The distance used in our approach does not take directionality into account. We opted for this strategy to make the method more globally applicable, as pointed out by the reviewer. We added this information in the corresponding method paragraph.

Given the rich set of data and processing, it is a pity that the authors do not enrich this work by analyzing in more detail their results. For instance, they could search for correlations and report if found or not between the range of the TF activity, the range of regulated gene changes and if that affects their predictability; frequently the so-called dual regulators might be activators with only negative autoregulation. Does autoregulation affect TF activity? Not all known metabolites were found, like the case of BetI for which choline is the known effector or for CysB whose effector is known (O-acetyl-L-serine (OAS)), as well as for AraC, MetR, FucR, among others with known effector. Any insight about those that failed?

Since the TF activities are directly derived from the regulated gene changes, the two variables are correlated by definition. Looking into how autoregulation affects TF activity is tricky because the TF's own gene is part of its own regulon and thus one cannot normalize the activity based on a TF's own expression without losing the activity signal. Typically, when a TF modifies its own expression, it is in response to an input signal. Thus, the signal within the expression of its own gene is also relevant to discover signal molecules, as done here. Additionally, TF's mRNA levels have been shown to be poor proxy of TF activity (PMID: 30462289). For more details, see answer on TF mRNA levels below.

Concerning the cases mentioned:

- CysB, BetI and MetR: their effectors were not detected by our metabolomics analysis.
- FucR: One of the three failed cases that we discussed in the main text. However, the metabolite next to its known effector was predicted as an input.
- For both FucR and AraC: these TFs are regulated by sugars, which have typically identical masses and are therefore not well distinguished by our metabolomics analysis. This point is discussed in the text for the Crp case, leading to signals not representing the abundance of individual sugars.

As discussed in the text, the main reasons of missing known effectors as are followed: The effector was not detected by mass spectrometry, the effector is not an activator of the TF activity (thus filtered out by our focus on activating interactions), the TF does not regulate enzymes (thus filtered out by the distance filtering), the TF had less 3 gene targets, low baseline activity or minimum absolute activity difference (thus filtered out at the TF activity inference step).

The value of publications is not limited to their data, but also to their reproducibility, as well as to their

ability to generate knowledge in a wider context and convey such knowledge to larger audiences, particularly in journals committed to large communities such as this one. I would like the authors to mention and even show the lack or amount of consistency between this final "TF activity" coming out of such computations where the known regulons are required, versus the direct quantification of differential mRNA TF levels from the transcriptomic datasets. The average correlation between mRNA levels and the corresponding proteins in bacteria is quite high as shown by the work of Terry Hwa. This might also offer the opportunity to utilize the distinction between TFs that bind in the conformation with the allosteric metabolite bound and those that bind in their absence.

As described in our answer to reviewer 2, taking TF expression into account is problematic due to autoregulating TFs. Additionally, while mRNA levels correlate relatively well with protein abundances in bacteria, it has been shown that TF expression is a poor proxy for TF activity (PMID: 30462289).

To illustrate this problem, we looked at the LeuO case. When using TF expression value instead of activity as suggested, we completely lose the association of LeuO with its validated signal molecule, with a correlation score near 0.

Nevertheless, we added a sentence commenting on this point in the newly added paragraph about TFs activity inference in the Discussion.

I strongly suggest that authors devote some thinking and write down a paragraph explaining what is different in VIPER for it to perform better than the other methods. Incidentally, in all graphs showing the comparative results of the different methods (Fig 1S), it is the VIPER lines that should have a clearly distinguishable color different from the other methods. Their current graph gives visual priority to the udt method in blue.

As asked by two reviewers, we added a text section on the description of the VIPER method and its advantages that could explain its higher performance in the Discussion. The use of analytic rank-based enrichment analysis as well as permutation tests for normalization makes VIPER highly robust and resilient to signature noise, regulon subsampling and sample quality, which might explain its superior performance on this dataset. As described below, we also extensively reworked Figure S1 to improve clarity.

It seems that the growth conditions of the transcriptomic analyses with those of metabolomics are not exactly the same or had some variations in the culture media. Please clarify.

The growth conditions have been matched to the best of our ability (same medium, strain, growth conditions, sampling timing). To ensure reproducibility, we extensively exchanged information with the actual researchers in the Palsson lab that generated the transcriptomics data. We therefore consider the conditions as comparable as possible and sufficiently identical for the conclusions reached. Nevertheless, inevitably some minute differences might still occur (different lab conditions, chemicals purity/property/age, recording of conditions not exact, ...). To compensate for such inevitable smaller differences, we also used a bootstrapping approach that limits the effect of potential poorly-matched individual condition.

In conclusion before minor details:

The problem being addressed is no doubt highly relevant. The strategy is not novel, but this work is a significant advance in the field given the comprehensiveness of its implementation. The paper is well written, with complex massive results well presented. However, as they recognize in the discussion, the limitation of the current implementation is the need for a large number of constraints to be satisfied: a combined large dataset of transcriptomics and metabolomics under the same growth conditions, and for a large number of growth conditions; transcriptomics should include KO of TFs for validation of the directional change in TF activity. Furthermore, previous knowledge of the set of target genes or regulons for the TFs whose allosteric metabolites are to be discovered is needed. Even if all these requirements are met, as in *E. coli* as here implemented, there is still the inherent difficulty of the fact that several

metabolites follow the same dynamical changes with TF activity. This will have no apparent solution in the near future as this is inherent in the biology of bacteria. (Q: Have you assessed whether precisely given these "dancing together" the regulatory systems should use only one metabolite as a signal of such metabolic cellular transition? How similar is the chemical structure of such co-dancing metabolites?). Finally, there is the technical current limitation of mass spec metabolites cannot be distinguished.

As pointed out by the reviewer and already discussed in the manuscript, the approach presented here has several limitations, most of which are technical in nature, and we can expect many of them to improve in the foreseeable future, for instance better mass spectrometry coverage of metabolites or a better/faster mapping of TF-gene pairs when moving to new organisms.

Indeed, correlating metabolites are often biological features that hamper direct identification of the actual signal, which should be at least one member in a correlating group. Nevertheless, there are ways to compensate for the lack of resolution, two of which have been used here: i) the distance filtering, which reduces the set of candidates to metabolites related functionally to the TF, and ii) measuring abundances over an even larger set of diverse conditions. Already in our dataset, by sampling over many diverse conditions, we reduced the chances of these paired metabolites to happen, in contrast to the Lempp et al paper that reported changes only between two growth conditions. In the worst case, correlating metabolites increase slightly the number of candidate metabolites, which would require more validation experiments. As pointed out in the discussion, the main limitation is currently the low throughput of laborious validation experiments and not really the hypothesis generation part of our approach. While our scientific focus was not on validation methods, this is an aspect on which throughput advances can be expected.

A completely different, not technical limitation is the availability of large transcript and metabolite data sets under identical conditions. For *E. coli* such data is available, and we show how to exploit it. For any new organism such data will have to be generated, but cost and throughput of transcriptomics and metabolomics experiments are continuously improving. Additionally, having TF mutant strains is not necessary to perform this analysis in a different organism. Here, we mostly used the TF mutant strains to assess the performances of different TF activity inference methods on a bacterial dataset.

We further commented on these points in the Results section and in the newly added Discussion paragraph on TFs activity inference.

Additional specific concerns and questions:

In line 76 the authors refer to Figure 1S as evidence to show that 34 out of 40 pairs were correctly assigned. But Fig 1S has 12 different panels, so it will be difficult for readers to identify where in this very rich figure to find the precise support for this statement. See also Fig 1 panel d with no distinctive description within the figure. Each panel in the figures should have a unique title to facilitate the understanding of the figures without the need to get deep into the footnote text.

We agree and we have modified Figure S1 accordingly. Specifically, we have simplified the figure by removing three uninformative panels, added panel titles, re-ordered and modified the legend accordingly to increase clarity, and added key informations both on the figure and in the text where cited. We have also now added the specific panels when citing Figure S1 to facilitate finding the correct data for each claim.

Line 300. Since the metabolomics analysis was performed on the extracellular medium without lysing the cells, is it possible that intracellular metabolites remain undetected, meaning that only external signals are being captured?

We only measured intracellular metabolites because those are the ones that TFs are expected to sense and interact with. Metabolite addition to the medium was only used to increase their intracellular concentration, thereby eliciting a signal.

We modified the corresponding Method section to further clarify this point.

Minor observations.

Line 321 "the E. coli 320 transcriptional regulatory network from RegulonDB v 10.5 curated from iModulonDB (Santos-Zavaleta et al, 2019)". This is a confusing sentence.

We modified the sentence to improve clarity.

The authors in the introduction make a good coverage of relevant citations, which nonetheless covers recent years, but very few to recognize for instance the very first characterization of allosterism in TFs.

We added citations to acknowledge early work on TF allosteric regulation in the Introduction.

Line 165-166: The sentence quoting ref (Santos-Zavaleta et al, 2019) is wrong.

We modified the sentence to improve clarity.

There was a typo in a footnote of a supplementary material, Figure S1: "..(h,i) Porportion of experimental conditions.."

Well spotted, we corrected it.

Reviewer #2:

In this work, the authors utilize a compendium of transcriptomics data (PRECISE 2.0) to correlate TRN signals (computed with VIPER + regulonDB) with metabolite levels across 40 conditions in order to hypothesize and validate new TF-metabolite interactions in E. coli. This work is next in a line of exciting endeavors to fill a major gap in our understanding of model organisms - our limited knowledge of small metabolite regulators of protein function, which until recently has been pieced together painstakingly through hypothesis-driven science. New methods are making it possible to search for these interactions en masse, which should have a transformative effect on systems biology both for data analysis and modeling. The approach of inferring TF activities from gene expression data has seen a resurgence in recent years and has shown itself to be quite successful in microbes; the missing piece has been metabolomics data which is still fairly limited in terms of publicly available datasets. As the authors fill this critical gap with metabolomics data for 279 metabolites over 40 conditions and experimentally validate their predictions, I applaud the effort. My primary concerns with the work are that I have some doubt about many of the predicted associations from their workflow (which could be alleviated through further analysis or experimental validation), and the presentation in terms of Figures, description of Results, and Methods could probably be improved. Specific comments are below.

Major Comments:

- The authors utilize a Spearman correlation between TF activities and metabolite abundances to assign putative regulatory roles. Would it make sense to account for the expression change of the regulator itself as well? For example, if a TF activity was constant, but the TF expression was down (due to some other regulator), and a metabolite level was up, that metabolite actually be an activator despite no direct association between activity and metabolite levels. Did the authors look at the role of TF expression itself on their predicted associations?

We considered accounting for the expression changes of the TFs themselves, as suggested by the reviewer. However, this is problematic because many TFs regulate their own expression in bacteria and thus the relationship between TF activity and TF expression is not the same between all TFs. Indeed, taking into account TF expression for autoregulating TFs would cancel out the actual signal in their TF activity. Moreover, due to the incompleteness of the mapped regulatory network, we do not know for all TFs whether they are regulating their own expression or not, making it impossible to account for it in a rule-based manner. Additionally, it has been shown that TF expression is a poor proxy for TF activity (PMID: 30462289). For these reasons, we did not take TF expression into account.

We explained this point as part of the newly added Discussion paragraph on TFs activity inference.

- Is there any evidence in the data of multiple metabolites regulating TFs? PurR is thought to be regulated by two transcription factors - I'm curious whether their approach based on Spearman correlations (i.e. assuming a single regulatory factor) could identify this or not. Related to this point, I understand that PurR was not identified as a statistical association - I'm also curious if for these 'failed' cases, whether any signal can be seen in the data that may have been below the top thresholds applied in the final predicted associations.

There are multiple cases where several metabolites were associated to a single TF, but without further experiments it is not possible to conclude whether they are all actual effectors or not. In the case of LeuO which we validated, one of the four predicted cases turned out to be an effector.

In theory it should be possible for this approach to capture several metabolite inputs, but it highly depends on the specific dynamics and direction of effect. It would also be restricted to activating interactions. More data is needed to confirm this, either by expanding the approach to more TF/metabolites, or validating more predictions.

Concerning known interactions, the vast majority of not recovered cases are explained by not detected metabolites or TFs that didn't pass the baseline and range activity thresholds. Either of these were excluded even before calculating correlations, hence they are not recoverable from "below the top thresholds".

- This is a matter of opinion but Figures 2-4 seem a bit drab and uninspiring, with key results somewhat difficult to glean. Inserting a few case studies e.g. on successful inference of known TF-metabolite interactions where we can see what the raw data looks like and how strongly the identified associations appear, would be helpful. Similarly, Fig 6 validates their predicted association but does not show the data for that association in the first place i.e. the expression of the LeuO regulon vs the metabolite level of the hypothesized effector. Figure 5 is the only figure that felt informative and interesting to read.

We agree and we have modified the figures to improve clarity and show more of the underlying data, with some improvements made to all main figures, including 8 newly added plots in Figure 2c, 4e, 5c-e and 6a. As suggested by the reviewer, we have notably added plots showing the distribution of correlation scores between specific TFs and all metabolites, highlighting their known or predicted effectors, for known cases that were both successfully recovered or not (Figure 4e), for predicted cases (Figure 5c,d,e) and for the case of LeuO (Figure 6a).

- It would be nice to compare their results to other high throughput experimental metabolite-protein interaction measurement methods (e.g. the recent Cell paper "Ligand interaction landscape of transcription factors and essential enzymes in *E. coli*" as well as the senior authors work such as <https://doi.org/10.1016/j.cell.2017.12.006>). Were the interactions that were inferred in this paper also detected in those works? Just trying to get a sense for the comparative discovery value of these approaches, and wondering if these studies could be a source of high-throughput validation of the predictions made in this work.

Comparison with Peng *et al.*, 2025 – Cell (PMID: 39862855): We made predictions for 42 TFs and Peng *et al.* for 46 TFs, 11 of which are present in both sets, including two that have a known activating signal. There is no overlap between our and their interaction predictions for these 11 TFs, but we correctly recovered both known activating signal molecules while they were not predicted by them. Notably, two of the 11 TFs also have known inactivating ligands (which are not covered by our approach), one of which was recovered by Peng *et al.*

Comparison with Lempp *et al.*, 2019 – Nature Communications (PMID: 31578326): We made predictions for 42 TFs and they for 30 TFs, 11 of which were present in both sets, including two that have a known activating signal. We correctly recovered one of the two known activating signal molecules while they were not predicted by them. Notably, three of the 11 TFs also have known inactivating ligands (which are not covered by our approach), one of which was recovered by Lempp *et al.* Additionally, two predictions were shared between the two sets: NanR with N-acetyl-D-glucosamine-P and DhaR with phosphoenolpyruvate.

We commented on these comparisons in the Discussion.

- On the choice of conditions for metabolomics data collection - Were these 40 conditions selected based on the 47 TF-associated conditions mentioned earlier? How much did the conditions seem to impact which TFs had a metabolite association (related to Figure 2a)? Do TFs activated and not activated make sense given the conditions chosen?

To select conditions, we first excluded conditions that were from laboratory evolution experiments, that used strains containing heterologous DNA, or undefined growth media. Then, we selected all conditions that we could reproduce with high confidence, i.e., for which we had the necessary equipment, chemicals and/or strains, and sufficient information for reproduction (for example exact chemical concentration was not available in some cases). This yielded the final set of “40 selected conditions” (described in detail in Table S2).

We investigated how well we recovered expectations of TFs activated or inactivated among the selected conditions, and the performance is overall similar to the results described in Figure 1 for all conditions. We now have included this analysis as a new panel in Figure 2 (Figure 2c), replacing the previous Figure 2b right panel which was redundant with the left panel, and commented on it in the main text.

- Around line 128 where prediction ability is discussed - can the authors plainly state what number of TFs they were able to successfully identify a known metabolite interaction from this approach (I believe the number is 5)? I understand thresholds apply but they can select some threshold and report Precision and Recall both if they like. At line 144 they state: "For positively correlating pairs, applying the stability and distance filters increased the proportion of recovered known interactions among predicted pairs about 100-fold, resulting in a final list of 103 TF-metabolite pairs, only 5 of which were previously known (Figure 5a, Table S7). Overall, this approach identified candidate input signals for 42 TFs." I must misunderstand something, because they say they increased the proportion of known associations 100-fold, and ended up with 5 known associations. Could the authors please clarify these statistics?

Yes, the final number of TFs for which we successfully identified a previously known interaction is 5. We further clarified this in the text after the different filtering are explained.

Here we meant the proportion of known interactions among the total number of pairs before and after the additional filtering steps. Before filtering, there were 9 known interactions among 16,657 pairs that passed the FPR-based correlation threshold in at least one of the 1,000 bootstrapping subsamples. After filtering, we retained a total of 103 pairs, which include 5 known cases, or an enrichment in known cases among pairs passing thresholds of about 100-fold.

We modified the corresponding sentence for clarity.

- I see in Figure 5a the known associations - e.g. Crp-cAMP that the authors discuss in the text. The authors comment on why these believe these associations appeared in their approach around line 148, but I don't understand this reduction to only 8 cases. They produced predicted associations for 42 TFs. They then reduce this to 18 TFs that had previously known small molecule associations. From these 18, they then seem to look only at 8 metabolites that are activators and measured in their study, if I understand correctly, and state they correctly inferred 5 of those, while the others known associated metabolites are nearby the predicted ones in the metabolic network. If I understand these filters correctly, why were only activating metabolites considered?

We focused on activating metabolites because of the better performance of our approach for this type of interaction, this is explained in the main text (lines 128-130, figure 4b).

- Related to the above comment - could the authors comment on why other well-known associations may not have been picked up? i.e. Cra-FDP, pyruvate with PdhR/IclR, ArgR-arginine and so on?

As discussed in the text, the main reasons of missing known effectors are as followed: The effector was not detected by mass spectrometry, the effector is not an activator of the TF activity (thus filtered out by our focus on activating interactions), the TF does not regulate enzymes (thus filtered out by the distance filtering), the TF had less than 3 gene targets, low baseline activity or minimum absolute activity difference (thus filtered out at the TF activity inference step).

For the specific cases highlighted by the reviewer:

- PdhR-pyruvate is an inactivating interaction
- Cra, IclR and ArgR were filtered out because of their low TF baseline activity

Since this was already stated in the text, we refrained from adding further text.

- What is the relevance of the metabolic network distance (Figure 4e)? They seem to use it as a criteria for selection of TF-metabolite interactions but I don't understand the principle. "more permissive choice to increase chances of discovering interactions involving metabolites also outside of the regulated enzymatic reactions." It seems like they did not want to include metabolites that could be affected indirectly through regulon perturbation (i.e. the TF affects the enzyme, which affects the metabolite, as opposed to the metabolite affecting the TF - but cases like arginine regulating ArgR would go against this strategy). I understand the challenge of disentangling cause and effect, I'm just not clear if applying a distance threshold is universally helpful. Could the authors comment on this?

The reasoning for the distance filtering is to focus on metabolites that are related to the regulated enzymes. That is because most known interactions are with metabolites that are substrate or product of enzymes directly regulated by the involved TF (Figure S2c). By filtering for these metabolites, we increase the performance of the approach in terms of recovery of known cases. The distance threshold used in our approach to yield the final list of prediction is 0, i.e., metabolites that are substrate or product of enzymes directly regulated by the involved TF, as shown in Figure S2.

The case of ArgR and arginine is thus not filtered out by the distance filtering. However, as mentioned above, ArgR was filtered out based on the minimum TF activity thresholds (Figure S1), which is why this pair is not present in our final list.

While responding to this point, we saw that the threshold written in the method section was wrong (and not matching to Fig S2) and we corrected it accordingly.

- Despite the care taken by the authors to optimize their workflow, it unclear how credible the predictions in Figure 5a are intended to be. Do we expect the SoxS is regulated by CTP/UTP and citrate/isocitrate? That Fnr is regulated by Threonine? These don't have the clear functional ties that their LeuO case has, at first glance at least. I'm sure that there is some kind of functional connection underlying many of these associations, but given the complexities of metabolic cause and effect - how do the authors propose we narrow in on the most likely associations to validate? It seems like LeuO was chosen as its predicted

effectors had an apparent topological logic to their putative regulatory role. The authors mention topologies in their discussion of Figure 5, but could the details here be presented on a case-by-case basis in Figure 5a as opposed to simple lump statistics in Figure 5b? Maybe this could be part of Supplementary Table 7 as well.

The strength of our approach is to rapidly narrow TF inputs down to a set of candidate metabolites. Indeed, for many the functional tie is not immediately obvious, although this does not imply that they are wrong. Within a predicted set, we do not expect all of them to be signals, but rather that the actual signal is amongst them. Some associations may also be indirect, as we have shown for two component systems where signals actually act on the kinase that regulates the TF.

The reviewer also asks whether we could propose a way to further narrow down on the most likely associations. Looking at the problem from a global perspective, we started from 48,267 possible pairs and narrowed down to 103 functional pairs. Given the available data, we have not found a way to narrow down further without decreasing the performance of this approach in terms of the recovery of known interactions. In our view the reduction in search space is already quite substantial and argue that all 103 pairs would be worthy of experimental follow ups.

In addition to the LeuO case, which we followed up on, we highlight three plausible examples in Figure 5, GadX, MngR and RutR. We have now added new plots showing the correlation profiles of these examples.

Concerning the topology in Figure 5, we added the information in Table S7 as suggested.

- The glutamine-gadX prediction is probably important enough to warrant experimental validation. A second validation case would go a long way as well toward addressing potential skepticism around many of the predicted interactions.

A good point and we did follow up on this prediction following the reviewer suggestion. However, the GadX protein proved very hard to purify and we failed to get any protein purified using the ASKA strains and different expression/purification protocols. Previous work focusing on GadX have used fusion proteins to study GadX (PMID: 11976288), fusing soluble proteins to GadX for purification, suggesting that this protein might be difficult to work with for *in vitro* studies.

This example illustrates well our point concerning the need for easier/faster validation approaches for these interactions, which is currently the main limitation in the field.

Minor Comments:

- Should the Introduction mention sigma factors as well or are these included in transcription factors? Not clear whether these were included, as their known regulators are not necessarily small molecules I believe, but they nevertheless greatly impact gene expression in bacteria.

Sigma factors were not included in this study. Indeed they are not typically known to sense small molecules.

To make it clear that they are not included, we added this information in the corresponding method paragraph.

- The Biorxiv manuscript for the PRECISE2.0 database is now published - it may be appropriate to update the link.

Done.

- Line 78 in the Results: "The PRECISE2.0 dataset includes 47 conditions related to 23 TFs," - Unclear what 'related' means or how these 47 conditions were selected from the entire dataset. Was this based on some activity threshold for a set of regulons that was examined?

We simply took every condition that had a TF known signal molecule added to the growth medium, there was no further selection. In total, there are 47 such conditions in the dataset, across which the known signal molecules of 23 TFs were added (some molecules are added to several conditions, some conditions have several molecules added, and some TFs have several known input molecules, hence the discrepancy between 47 conditions and 23 TFs).

We modified the sentence to make that point clearer.

- Line 79 in the Results: "In 83% of these cases," Some discussion of the minority of cases that did not work would be warranted, as well as how the 'expected' direction of perturbation was identified. Was this straightforward or were there any ambiguous cases? This paragraph is described in such a general way that the results seem a bit vague and less convincing than necessary (despite the strong signal seen in the figure).

Here, by expected direction we meant that the activity of a TF should decrease when that TF is mutated. We modified to corresponding sentence to make that point clearer.

Additionally, we added a sentence describing the minority of cases that did not work. In these cases, the activity changes were relatively low, suggesting little effect of the addition of the metabolite, which could be due low metabolite intake, fast metabolite degradation, or to the corresponding TF not being active/expressed in the specific condition.

- Line 121 - How necessary was this bootstrapping approach? What kinds of problematic issues arose when this step was originally excluded? Per a more technical question I had further below, I had a hard time understanding the rationale here and the problem that this approach solves. What is a "poorly matched growth condition" for example?

This approach was used for two reasons:

1. Reduce the chances that an association is made solely because of a single condition is driving the correlation
2. Reduce the effect of potentially poorly matched growth condition

By "poorly matched growth conditions" we mean conditions that, despite our best efforts to match the experimental conditions of the Palsson lab, was done differently between the two labs and might lead to individual points biasing the correlation.

We modified the corresponding sentence to make that point clearer, as well as a sentence in the method section to expand on the reasoning.

- Line 223 - 41 TFs are mentioned here, but 42 were mentioned earlier in line 147 (later 41+Crp were mentioned, so maybe 42 is the correct number?).

Here, we meant the set excluding Crp, which is our final set of high-confidence interactions, since Crp's results are highly biased by its regulon size and indistinguishable sugars.

- Could Figure 2a be somehow organized to impart more information? TFs could be sorted by regulon size, organized by functional category or at least sorted by range in activity.

We have modified Figure 2a so that it is sorted by activity range.

- Fig 3A - PCA plots typically show the explained variance along the X and Y axis. A biplot would be even more informative, to show the top metabolites driving separation in the top 2 PCs.

We added explained variance information on both PCA plots (Figures 1 & 3).

- Since VIPER was chosen as the TF activity inference method based on performance evaluated in Figure S1 - some brief introduction into how it works would be warranted.

As asked by several reviewers, we added a text section describing the VIPER method and its advantages in the Discussion.

- Line 323 of the Methods is not clearly written to me: "To account for incompleteness of the regulatory network, the network was subsampled, as previously described (Ortmayr et al, 2019), for each TF within ten subnetworks containing 40 randomly chosen additional TFs for which activities were inferred. For each TF, the median of all subsampling simulations was calculated and used as final activity value." I'm having a hard time understanding what was done here, why the network needs to be subsampled, or what it means for "the median of all subsampling simulations" to be calculated (presumably this means the median of the decoupleR-computed activities across different subsampled regulons?).

We are using a subsampling approach to account for the incompleteness of the regulatory network. Indeed, we expect that our current knowledge of the regulatory network is missing regulatory interactions, which are nonetheless affecting gene expression in a way that we cannot account for. To minimize the effect of such missing information, each TF activity is inferred within a series of local subnetworks where it is evaluated alongside a randomly selected group of other TFs.

Subsampling the TRN helps to avoid biased or unstable TF activity inference caused by uneven regulon sizes, missing data or condition-specific interactions. By iteratively inferring activity in smaller, randomly composed subnetworks and aggregating the results using the median value, we get more robust estimates of TF activity across the whole network.

We added a sentence on this point in the corresponding method paragraph.

- Similarly, line 335 of the Methods was also confusing to me: "Based on accuracy on the direction metric, all TFs were filtered on baseline activity and minimum absolute activity difference as thresholding on these criteria showed improved performance on correctly assigned TF activities. To that aim, thresholds were chosen as minimizing the X2 p-value when dynamically thresholding across the range of corresponding values for increased correctly assigned TF activities." Could this description be expanded and stated in a simpler way to make it easier to understand what was done? Multiple thresholds/filters were discussed but it's not clear what values were chosen or why/how.

We found that the TF inference method performs poorly for TFs that always have very low activity, or activity that almost never varies. We have thus implemented thresholds on these two aspects to exclude the corresponding TFs. We have modified the text to make that clearer as well as extensively modified Figure S1. Specifically, we have simplified the figure by removing three uninformative panels, added panel titles, re-ordered and modified the legend accordingly to increase clarity, and added key informations both on the figure and in the text where cited. We have also now added the specific panels when citing Figure S1 to facilitate finding the correct data for each claim.

Reviewer #3:

This study by Sauer and colleagues uses available transcriptomic datasets with high-throughput metabolomics to identify transcription factor-metabolite interactions that could impinge on transcription factor activity. Through an intricate analyses, the study identifies 76 novel metabolite-TF interactions and the data is validated by examining the predicted input signal for LeuO. Overall, this exhaustive study provides a useful resource to investigate various metabolism-dependent gene-regulatory networks in *E. coli* and the approach could very well be used to study similar networks in other bacteria.

However, the following concerns need to be addressed to further strengthen this study:

Line 77 to 81: The expression of genes corresponding to targeted TFs were checked in the presence of specific metabolites. For this analysis, the techniques and conditions used to quantify gene expression is not clear. It would be useful if the parameters are mentioned in the methods.

For this analysis we used published RNA-seq data from the PRECISE2.0 public dataset. We modified this part of the Results section to increase clarity.

Line 80: In figure 1D, the authors have shown the change TF activity when a metabolite is added in the growth medium, compared to a paired control condition. However, Figure1D legend says "activity difference between the TF knockout mutant and the corresponding wild-type control conditions". It is difficult to interpret the data from the text provided in result section vs corresponding figure legend. Authors should provide enough information or rewrite the result section and figure legends for clear understanding.

Indeed, there was a mistake in the legend of Figure 1d. As described in the text, Figure 1d corresponds to cases where the known signal molecule of a TF has been added to the medium, not to TF mutants. We corrected the legend text accordingly.

Line 107 and 108: "The metabolites were extracted at the exponential phase, where, the cells were harvested at OD600 reported for RNA extraction" (as per methods, line 298-299). However, considering the diversity in conditions and strains used in this study, the growth in these conditions could vary across samples. Further, this will eventually lead to variation in time required to attain exponential phase and OD at exponential phase. It is not clear how the authors manage to identify the correct timepoint for extraction, especially when both of above factors vary largely across the samples.

Indeed, there were differences in the timing to reach exponential phase between the growth conditions. To extract at mid exponential phase for each sample, we did two things:

- All conditions were first grown to stationary phase to estimate the time of sampling.
- To ensure reproducibility, we discussed extensively with the actual researchers from the Palsson lab who generated the transcriptomics data. Part of this exchange was detailed timing/OD information to match their experiments.

With this information, we then sampled each condition at their determined time of sampling (~ 2 to 5h post inoculation). We added this information in the method section.

The difference in metabolite quantification could also be contributed by the difference in extraction efficiency. Was this taken into consideration? If so, a line on how this was done will be useful.

It is indeed an inevitable feature of metabolomics that not all metabolites are extracted with the same efficiency. However, for each metabolite the extraction efficiency remains identical across different samples, hence it does not affect the results.

Line 189-190: Out of 4 candidate inputs for LeuO effect of only one metabolite is validated. For other three candidates, it is possible that derivatives of, or chemical modifications (acetylation, lactylation etc.) caused by, input metabolites could influence TF activity. Is it possible to identify such cases from the existing data?

The four candidates found for LeuO are linked because they are from the same metabolic pathway. That is probably why they all come up as hits, and also why functional validation is required. There is no reason to expect derivatives of these molecules to be signals.

Figure S4: The gel shift assay shows a very faint band on addition of 2-isopropylmalate and is difficult to

interpret.

Indeed, the effect detected on the EMSA is mild, which is not rare for these types of interactions and could be explained by sub-optimal *in vitro* conditions. While mild, this effect was consistently seen across all replicate experiments and is statistically significant.

It is not an absolute requirement but an in-vivo validation of TF-metabolite interaction would provide more impact to the work.

Here, we focused on developing a pipeline for the high-throughput identification of these interactions, with the main result of providing evidence for 103 new metabolite input signals. While we agree that the detailed characterization of these interactions and their effect in vivo is important and is an obvious next step, this is very much a community effort that we believe to be beyond the scope of this work. Nevertheless, we attempted to follow up on the GadX predictions, which failed due to technical difficulties in purifying the GadX protein, as explained in our answer to reviewer 2 above.

Line 115-116: Figure reference is missing.

This has been corrected.

Line 125: Shouldn't Fig. 4B be Fig. 4C?

This has been corrected.

Reference missing for line 298-299.

This has been corrected.

Supplementary figure: S1H and S1I: x-axis is not labelled.

This has been corrected.

26th Jun 2025

Manuscript Number: MSB-2025-12935R

Title: Predicting input signals of transcription factors in Escherichia coli

Author: Julian Trouillon

Alexandra Huber

Yannik Trabesinger

Uwe Sauer

Dear Uwe,

Thank you for sending us your revised manuscript. We have now received feedback from the three reviewers who evaluated your study. As you will see below, Reviewers #2 and #3 are satisfied with the performed revisions, while Reviewer #1 has mentioned a few issues. We kindly ask that you respond to these comments and address the issues raised in writing, as appropriate.

In addition, we would ask you to address some remaining issues listed below:

1. Please remove the Authors' contribution section from the manuscript file.
2. Please provide up to five keywords.
3. Add the missing funding information-ETH Career Seed award (ETH Zürich)-to the submission system.
4. "Disclosure statement and competing interests" should be renamed to "DISCLOSURE AND COMPETINGINTERESTS STATEMENT". Please also add a sentence: "US is a member of the Advisory Editorial Board of Molecular Systems Biology. This has no bearing on the editorial consideration of this article for publication."
5. Tables EV1-7: As these tables are quite large, please update their designation from Tables EV1-EV7 to Datasets EV1-EV7. Accordingly, please revise the source file names, titles, legends, and all manuscript callouts to reflect this change. Additionally, legends should be removed from the manuscript file and instead included as a separate tab or sheet within each corresponding Excel file.

Table EV8 should be renamed to Table EV1 with the corresponding callout; legend should be removed from manuscript file and included above the table in the Excel file

6. Please ensure that all table and figure callouts appear in sequential order. Currently, there is a callout for Table S3, but this table has not been uploaded. Kindly review and resolve this discrepancy.

7. Source data: Source data files should be organized in a clear structure: one folder per figure. Each folder should then be compressed and uploaded as a separate .zip file.

For example, all source data for Figure 1 should be saved in a single folder and uploaded as "SD Figure 1.zip".

8. Please provide a "standfirst text" summarizing the study in one or two sentences (approximately 250 characters, including space), three to four "bullet points" highlighting the main findings and a "synopsis image" (550px width and 400-600 px height, PNG format) to highlight the paper on our homepage.

Here are a couple of examples:

<https://www.embopress.org/doi/10.15252/msb.20199356>

<https://www.embopress.org/doi/10.15252/msb.20209475>

<https://www.embopress.org/doi/10.15252/msb.209495>

9. Please address the following issues related to figure legends:

- Please note that the legend for figure EV1 is not provided in the sequential manner (legend for figure EV1 F is provided before legend of figure EV1 E). This needs to be rectified.
- Please note that the exact p values are not provided in the legends of figures EV1 A, B
- Please indicate the statistical test used for data analysis in the legends of figures 6B, C; EV1 D, F
- Please note that the box plots need to be defined in terms of minima, maxima, centre, bounds of box and whiskers, and percentile in the legends of figures 1C, D; 2B, C; 6B, C
- Please note that information related to n is missing in the legends of figures 1D, 2B, C; 3C, EV1 A-C

10. Please ensure that the manuscript sections are presented in the following order: Title page - Abstract - Keywords -

Introduction - Results - Discussion - Methods - Data Availability - Acknowledgements - Disclosure and Competing Interests Statement - References - Figure Legends - Table(s) - Expanded View Figure Legends.

Please resubmit your revised manuscript online, with a covering letter listing amendments and responses to each point raised by the referees. Please resubmit the paper ****within one month**** and ideally as soon as possible.

When you resubmit your manuscript, please download our CHECKLIST (<https://bit.ly/EMBOPressAuthorChecklist>) and include the completed form in your submission. *Please note* that the Author Checklist will be published alongside the paper as part of the transparent process (<https://www.embopress.org/page/journal/17444292/authorguide#transparentprocess>)

Click on the link below to submit your revised paper.

Kind regards,
Jingyi

Jingyi Hou, PhD
Senior Editor
Molecular Systems Biology

*** PLEASE NOTE *** As part of the EMBO Press transparent editorial process initiative (see our Editorial at <https://dx.doi.org/10.1038/msb.2010.72> , Molecular Systems Biology will publish online a Review Process File to accompany accepted manuscripts. When preparing your letter of response, please be aware that in the event of acceptance, your cover letter/point-by-point document will be included as part of this File, which will be available to the scientific community. More information about this initiative is available in our Instructions to Authors. If you have any questions about this initiative, please contact the editorial office (msb@embo.org).

Reviewer #1:

In general, the point-by-point response was much more effective given the considerable number of questions, and my appreciation is that the revised version missed addressing several of the concerns. For instance, in my case, I emphasized that this approach it not new in its strategy, nonetheless this observation did not affect their new version, which requires careful reading to identify it, because the authors did not say so clearly. The relevance of this work is the implementation of a workflow directed to identify ligands that bind TFs using omics data, given the relevance of this knowledge. However, given the complexity of the biology (i.e. autoregulation of TFs, similar dynamics of several metabolites, etc) and the need for omics data, the validity of their predictions in the absence of final experimental validation remains limited. This is why, the concern also mentioned by another reviewer is the lack of an assessment of the ranking or probability of the different predictions.

Reviewer #2:

I have reviewed the manuscript changes and responses, and the authors have sufficiently addressed my previous concerns.

Reviewer #3:

The authors have satisfactorily addressed the major concerns raised, and accordingly have made changes in the revised manuscript. This study reports an impactful work wherein a pipeline for high-throughput identification of transcription factor-metabolite interactions has been developed. Altogether, this work advances the current knowledge and could serve as a potential resource for related studies.

Reviewer #1:

In general, the point-by-point response was much more effective given the considerable number of questions, and my appreciation is that the revised version missed addressing several of the concerns. For instance, in my case, I emphasized that this approach is not new in its strategy, nonetheless this observation did not affect their new version, which requires careful reading to identify it, because the authors did not say so clearly. The relevance of this work is the implementation of a workflow directed to identify ligands that bind TFs using omics data, given the relevance of this knowledge. However, given the complexity of the biology (i.e. autoregulation of TFs, similar dynamics of several metabolites, etc) and the need for omics data, the validity of their predictions in the absence of final experimental validation remains limited. This is why, the concern also mentioned by another reviewer is the lack of an assessment of the ranking or probability of the different predictions.

In the last paragraph of the introduction, we state that the correlation approach has been used before, cite the corresponding papers, and explain what it was applied to in these previous works and what we are doing with it in the present paper (lines 55-62). As requested by the reviewer, we have now added an additional sentence in the Discussion to clearly state that the correlation approach that we are using has been used before (lines 272-274).

As much as we agree with the reviewer's wish that for any type of hypothesis generating discovery method one would like to have many validation experiments, we respectfully want to point out that i) our method predicted a quite restricted set of 80 interactions from almost 50'000 possible ones and ii) we provided strong evidence that true interactions are highly enriched among our predictions. Due to the labour-intensive nature of these validations, we focused on one novel case (LeuO). To go further is, in our view, beyond the scope of this paper. We believe that our results are a very helpful starting point for the community to focus their future studies.

Concerning the ranking or probability of different predictions: As stated in our previous answers to reviewers, we have not found a way to narrow down or prioritize further among the set of predicted interactions. This means that we consider all predictions equally worthy of further investigation. For example, giving priority based on higher correlation scores among the pairs that already passed the correlation threshold would decrease the performance of our approach in terms of true positive recovery. In several cases, the true interaction was not necessarily the top 1 correlation but rather was among the top few that passed our thresholds, as shown on some of the new plots that we added as part of our response to the previous round. We have added a sentence commenting on this point in the corresponding results paragraph (lines 157-159).

Reviewer #2:

I have reviewed the manuscript changes and responses, and the authors have sufficiently addressed my previous concerns.

Reviewer #3:

The authors have satisfactorily addressed the major concerns raised, and accordingly have made changes in the revised manuscript. This study reports an impactful work wherein a pipeline for high-throughput identification of transcription factor-metabolite interactions has been developed. Altogether, this work advances the current knowledge and could serve as a potential resource for related studies.

30th Jun 2025

Manuscript number: MSB-2025-12935RR

Title: Predicting input signals of transcription factors in Escherichia coli

Dear Uwe,

Thank you again for sending us your revised manuscript. We are now satisfied with the modifications made and I am pleased to inform you that your paper has been accepted for publication.

Kind regards,
Jingyi

Jingyi Hou, PhD
Senior Editor
Molecular Systems Biology
